# BEYOND RE-BALANCING: DISTRIBUTIONALLY ROBUST AUGMENTATION AGAINST CLASS-CONDITIONAL DISTRIBUTION SHIFT IN LONG-TAILED RECOGNITION

## ABSTRACT

As a fundamental and practical problem, long-tailed recognition has drawn burning attention. In this paper, we investigate an essential but rarely noticed issue in long-tailed recognition, Class-Conditional Distribution (CCD) shift due to scarce instances, which exhibits a significant discrepancy between the empirical CCDs for training and test data, especially for tail classes. We observe empirical evidence that the shift is a key factor that limits the performance of existing long-tailed learning methods, and provide novel understanding of these methods in the course of our analysis. Motivated by this, we propose an adaptive data augmentation method, Distributionally Robust Augmentation (DRA), to learn models more robust to CCD shift. A new generalization bound under mild conditions shows the objective of DRA bounds balanced risk on test distribubtion partially. Experimental results verify that DRA outperforms related data augmentation methods without extra training cost and significantly improves the performance of some existing long-tailed recognition methods.

## 1 INTRODUCTION

Recently, visual recognition has achieved significant progress, driven by the development of deep neural networks (He et al., 2016) as well as large-scale datasets (Russakovsky et al., 2015). However, in contrast with manually balanced datasets, real-world data often has a long-tailed distribution over classes i.e. a few classes contain many instances (head classes), whereas most classes contain only a few instances (tail classes) (Liu et al., 2019; Van Horn & Perona, 2017). Training models on long-tailed datasets usually leads to degenerated results, including over preference to head classes, undesired estimation bias and poor generalization (Zhou et al., 2020; Cao et al., 2019; Kang et al., 2019).

To solve above issues, various solutions have been proposed. Many of them focus on addressing imbalanced label distribution for simulating class-balanced model training. Direct re-balancing, like re-sampling and re-weighting, is the most intuitive (Huang et al., 2016; Zhang et al., 2021b). Recently, the two-stage methods, which apply re-balancing strategy in tuning classifier (Kang et al., 2019) or defer re-weighting after initialization (Cao et al., 2019), have been verified effective. Logit adjustment uses margin-based loss or post-hoc adjustment to rectify the biased prediction caused by long-tailed distribution (Menon et al., 2020; Ren et al., 2020; Hong et al., 2021). Formally, denoting an input-label pair as $(x, y)$, classification or recognition models are trained to estimate the posterior probability $P(y|x) \propto P(y)P(x|y)$. In long-tailed recognition scenarios, most solutions actually obey the following assumption: the class distribution $P(y)$ shifts from training to test (usually class-imbalanced in training but class-balanced in test), while the class-conditional distribution (CCD) $P(x|y)$ keeps consistent, i.e. $P_{train}(y) \neq P_{test}(y)$ and $P_{train}(x|y) = P_{test}(x|y)$ (Menon et al., 2020; Ren et al., 2020). Under this assumption, a series of methods including direct re-balancing and logit adjustment have been proved Fisher-consistent (Menon et al., 2020).

We argue that although the consistent CCD assumption(Menon et al., 2020) is reasonable if there is no sampling bias within each class, estimating $P(x|y)$ by empirical CCD is unreliable, especially for tail classes where the samples are extremely scarce. Therefore, to obtain a generalizable model, the shift between empirical CCD and the ideal CCD cannot be ignored. Our focus does not overlap

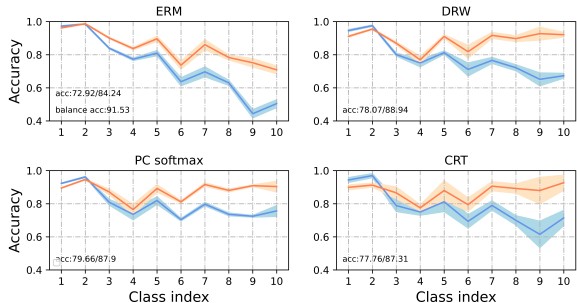

Figure 1: Accuracy on CIFAR10-LT with or without removing CCD shift. All methods show significant improvement after removing shifts. And the improvement mainly appears in classes with fewer instances, which verifies the empirical CCD distributions of tail classes are more unreliable. Shaded regions show 95% CIs over 5 runs.

with existing methods that attend to scarce tail instances or inconsistent $P(x|y)$. Transfer learning and data augmentation have been proven effective from the motivation of increasing the diversity of tail classes (Kim et al., 2020; Zhong et al., 2021; Zhou et al., 2022). But they are still possibly biased due to unreliable empirical distribution and usually lack theoretical guarantee. Some recent works focus on inconsistent class-conditional distribution caused by domain bias or attribute-wise imbalance (Gu et al., 2022; Tang et al., 2022), which is under shift from unreliable estimation as well. Nevertheless, the influence of CCD shift has not been thoroughly investigated and there has been no effective solution yet.

In this work, we perform an empirical study to quantify the effect that the shift of $P(x|y)$ will have on long-tailed recognition, by using CCD from balanced datasets as an oracle to alleviate CCD shift. With this oracle, the performance of existing methods is significantly improved, as shown in Figure 1, which indicates CCD shift is a key factor in limiting the performance of long-tailed learning. From the CCD shift perspective, we also give new insights to counter-intuitive facts of previous methods e.g. why decoupling methods (Kang et al., 2019) work and how Fisher-consistent parameter in logit adjustment (Menon et al., 2020) gets sub-optimal performance.

Motivated by our empirical study, to enhance robustness against CCD shift, we propose Distributionally Robust Augmentation (DRA) which assigns class-aware robustness, generalizing Sinha et al. (2017) and admits a novel generalization bound. This bound verifies models robust to CCD shift benefit long-tailed recognition. Our experiments show DRA improves various existing methods significantly, validating our theoretical insights. Our main contributions are highlighted as follows:

- We identify a rarely noticed but essential issue in long-tailed recognition, class-conditional distribution (CCD) shift, and provide new insights from the CCD shift view into some existing methods.

- To train models robust to CCD shift, we propose DRA with theoretically sound modifications over prior DRO methods(Sinha et al., 2017), which admits a novel generalization bound verifying that models robust to CCD shift benefit long-tailed recognition.

- Extensive experiments on long-tailed recognition show effectiveness of DRA: it significantly improves existing re-balancing methods and achieves comparable performance to state-of-the-arts. Moreover, DRA outperforms related data augmentation methods without additional training costs.

## 2 RELATED WORK

### 2.1 LONG-TAILED RECOGNITION

**Re-balancing methods.** In this section, we review a broader scope of re-balancing methods. We regard methods trying to solve the prior shift of $P(y)$ as re-balancing methods. Long-tailed learning can be considered as a special label shift problem whose testing label distribution is known while the

original label shift problem needs to detect and estimate the unknown label distribution (Lipton et al., 2018; Garg et al., 2020; Storkey, 2009; Latinne et al., 2001). Directly re-sampling or re-weighting by the inverse of class frequencies is intuitive but causes over-fitting to tail classes or optimization difficulties (Zhang et al., 2021b; Cui et al., 2019). Deferring use of re-balancing strategies (e.g. DRS/DRW (Cao et al., 2019; Zhou et al., 2020)) and decoupled methods which only re-balance the classifier (e.g. CRT, LWS (Kang et al., 2019)) are proved more effective (Zhong et al., 2021; Zhang et al., 2021a). Beyond class frequencies, more efficient coefficients for re-balance are explored. Focal (Lin et al., 2017) and CB (Cui et al., 2019) use the predicted logits or the effective numbers to re-weight classes, while IB (Park et al., 2021) adopts influence function of every instance to revise weights. Another strand of re-balancing methods proposes margin-based loss to heuristically assign a larger margin to tail classes, e.g. LDAM (Cao et al., 2019) and RoBal (Wu et al., 2021). Logit adjustment (Menon et al., 2020) utilizes margin-based loss to adjust logits, and the proposed logit adjustment loss and post-hoc adjustment are equivalent to PC softmax and Balanced softmax respectively (Hong et al., 2021; Ren et al., 2020). As for the convergence of re-balancing methods, it has been theoretically proved with re-balance weight matching the margin, the loss is Fisher-consistent under consistent class-conditional distribution assumption (Menon et al., 2020). Recent works find logit adjustment leaves space to be improved by adding an instance-aware confidence function (Zhong et al., 2021) or regularizer on posterior probability (Hong et al., 2021).

**Information transfer and data augmentation.** Some methods use transfer learning from head to tail classes to remedy the lack of data diversity. M2m (Kim et al., 2020) formulates an additional optimization to translate head instances to tail instances and GIT (Zhou et al., 2022) uses GAN (Goodfellow et al., 2014) to learn nuisance transformations to transfer invariances for tail classes. A recent assumption is that features from head classes and tail classes share intra-class variances while observed variances positively correlate with class frequencies (Yin et al., 2019). Under this assumption, some methods transfer variances by sharing classifier weights (Liu et al., 2021) or sampling features from estimated Gaussian distributions (Liu et al., 2020). On the other hand, data augmentation methods are also proved effective to increase the diversity of instances, e.g. ISDA (Wang et al., 2019), mixup (Zhang et al., 2017). To adapt to long-tailed settings, some methods make modifications by adjusting the probability of classes in mixup (Chou et al., 2020; Xu et al., 2021) or adding label smoothing after mixup training (Zhong et al., 2021), while MetaSAug (Li et al., 2021) uses meta-learning to estimate covariance in ISDA.

**Challenging $P_{train}(x|y) = P_{test}(x|y)$.** Very recent works challenge the shift of $P(x|y)$ as well and propose new perspectives for the long-tailed recognition problem. Tang et al. (2022) discover that real-world data exhibit both class-wise imbalance and attribute-wise imbalance, but the latter has long been overlooked. So they propose invariant feature learning (IFL) framework using IRM (Ahuja et al., 2020) to deal with this generalized long-tailed learning problem. Besides, Gu et al. (2022) considers the situation that the instances of a class are from different domains, and proposes a united framework based on meta-learning under the multi-domain setting. GIT (Zhou et al., 2022) implicitly challenges $P_{train}(x|y) = P_{test}(x|y)$ with the discovery that the model cannot give invariant predictions for class-agnostic transformations on tail classes because of the less diversity of tail classes.

**Other related methods.** DRO-LT (Samuel & Chechik, 2021) proposes a margin-based loss for a better feature extractor which is an upper bound of a Distributionally Robust Optimization problem. CDT (Ye et al., 2020) finds the feature deviation between the training and test data, especially for minor classes, and proposes class-aware temperature. MFW (Ye et al., 2021) further explains feature deviation is from larger gradients on tail classes in the training process and proposes to mix features within mini-batches to balance the gradients.

## 2.2 DISTRIBUTONALLY ROBUST OPTIMIZATION

Distributionally Robust Optimization (DRO) (Kuhn et al., 2019; Delage & Ye, 2010) considers the worst-case risk minimization on a series of potential distributions, called uncertainty sets, which are usually determined based on various computations, e.g. Wasserstein distance Kuhn et al. (2019), f-divergence Namkoong & Duchi (2016) and moments constraints (Delage & Ye, 2010). WRM (Sinha et al., 2017) gives a general solution to the below DRO problem for smooth loss functions:

$$\min_{\theta} \sup_{\hat{p} \in \mathcal{P}} E_p[l_\theta(x,y)], \mathcal{P} = \{\hat{p}|W_c(\hat{p}, p_N) < r\},$$

where the uncertainty set is a ball under Wasserstein distance $W_c$. WRM converts the Lagrange penalty of above problem to a min-max optimization with a constant Lagrange multiplier $\lambda$ and solves it by Algorithm 2:

$$\min_{\lambda, \theta} \sup_{(x', y')} E_{(x,y) \sim p_N}[l_\theta(x', y') - \lambda c((x', y'), (x, y))]$$

Our DRA generalizes WRM by using class-wise uncertainty set and generating a sequence of examples in the inner-optimization, which shows superiority over WRM theoretically and empirically. Zhang et al. (2020b) utilizes DRO to train models which are robust to label shift, orthogonal to our model which is robust to potential CCD shift. Lin et al. (2022) surveys recent studies on DRO.

## 3 CLASS-CONDITIONAL DISTRIBUTION SHIFT IN LONG-TAILED RECOGNITION

In this section, we give empirical studies on unreliable class-conditional distribution (CCD) estimation in long-tailed recognition. We first introduce problem setup. Then we leverage balanced distribution to manually remove CCD shift in ablation experiments. Finally, we present theoretical and empirical analysis based on the experimental results.

### 3.1 PROBLEM SETUP

In classification/recognition setting, a labeled instance is a pair $(x, y)$, where $y$ takes value in $[L] \doteq \{1, ..., L\}$ with $L$ the number of classes and $x \in \mathbb{R}^m$. We consider a $\theta$-parameterized classifier $f$, e.g. a neural network, which outputs $f_\theta(x) \in \mathbb{R}^L$ to estimate $P(y|x)$. Denote training set as $\{(x_i, y_i)\}_{i=1}^N \sim P_{train}$, with $P_{train} = \sum_{j \in [L]} P(y_j) P(x|y_j)$ and $N_j$ the number of instances from class $j$. In the long-tail scenario, $P(y)$ is imbalanced, and we assume that $P(y_1) > ... > P(y_L)$. In practice, the model is desired to perform well on all classes for the purpose of e.g. detecting rare species or making fair decisions (Van Horn & Perona, 2017; Hinnefeld et al., 2018). Hence, the distribution to evaluate is usually class-balanced i.e. $P_{test} = \frac{1}{L} \sum_{j \in [L]} P(x|y_j)$, In other words, the goal is minimizing the balanced risk (Menon et al., 2020)

$$R_{bal} = \frac{1}{L} \sum_{j \in [L]} P_{x|y_j}(y_j \neq \arg\max_{y \in [L]} \{f_\theta(x)\}_y) \tag{1}$$

$$= \frac{1}{L} \sum_{j \in [L]} E_{P(x|y_j)} \{l_{0/1}(y_j, \arg\max_{y \in [L]} \{f_\theta(x)\}_y)\} \tag{2}$$

In previous work, to estimate $R_{bal}$, $P(x|y_j)$ is approximated by a loss function $l$ e.g. cross-entropy and the empirical distribution (Menon et al., 2020; Vapnik, 1991) $P_{N_j}(x|y_j) = \frac{1}{N_j} \sum_{i:y_i=y_j} \delta(x_i, y_i)$, which is unreliable under long-tailed setting due to scarce instances of tail classes. It can be explained more formally by the following proposition.

**Proposition 1** *For a $L$-Lipschitz loss function $l$, $P(x|y)$ is a conditional distribution on $\mathbb{R}^m$ and $P_N(x|y)$ is the empirical distribution of $P(x|y)$ estimated from $N$ instances. Denote $A \doteq E_{P(x|y)}\{\|x\|^\alpha\} < \infty$ for some $\alpha > 0$, then $\exists\ c_1, c_2$, depending on $A$, $\alpha$, we have $\left| E_{P(x|y)}\{l(x, y)\} - E_{P_N(x|y)}\{l(x, y)\} \right| < t$ with probility at least $1 - c_1 / e^{(\frac{t}{L})^{min\{m,a\}} c_2 N}$.*

It says with very few instances e.g. tail classes, training performance does not ensure generalization, i.e. the shift between empirical distribution and true distribution cannot be ignored. That is the so-called CCD shift. The proof of this proposition can be found in Appendix A. Unfortunately, unreliability of CCD cannot be directly estimated from the bound of Proposition 1 due to unavailable real distribution.

### 3.2 IDENTIFYING CCD SHIFT BY LEVERAGING BALANCED DISTRIBUTION

Many benchmarks of long-tail recognition, such as CIFAR10/100-LT (Cao et al., 2019) and ImageNet-LT (Liu et al., 2019), are down-sampled from balanced datasets. Formally, instances

Table 1: Accuracy of decoupling method CRT on CIFAR10-LT under different settings without or with removing shift sampling. The underline means accuracy of imbalanced features and small font denotes difference between re-balanced and imbalanced features. More results are in Table 4.

| feature re-balance | classifier adjust | base | removing shift |
|---|---|---|---|
| - | CRT | 77.72 | 87.07 |
| DRW | CRT | 75.97$_{-1.75}$ | 88.42$_{+1.35}$ |
| DRW | - | 78.06 | 89.01 |

from balanced datasets and long-tailed dataset are sampled respectively from

$$P_{N,bal}(x,y) = \sum_{j \in [L]} P_{bal}(y_j) P_{N_j,bal}(x|y_j), \tag{3}$$

$$P_{N,LT}(x,y) = \sum_{j \in [L]} P_{LT}(y_j) P_{N_j,LT}(x|y_j). \tag{4}$$

Here $P_{N_j,bal}(x|y_j) = \frac{1}{N_{bal,j}} \sum_{\{i:y_i=y_j\}} \delta(x_i, y_i)$ and $P_{N_j,LT}(x|y_j) = \frac{1}{N_{LT,j}} \sum_{\{i:y_i=y_j\}} \delta(x_i, y_i)$. Besides, $P_{bal}(y)$ is uniform and $P_{LT}(y)$ is imbalanced. To conduct an ablation comparison with respect to CCD, we replace the CCD in (4) with $P_{N_j,bal}(x|y_j)$, which is more reliable by Proposition 1, as an oracle for real CCD. While $P_{N_j,bal}(x|y_j)$ may not approximate real CCD well, as long as the approximation brings significant improvement on performance (as in our experiments and see more in Appendix B.1), we can identify CCD shift according to Proposition 1. That is, sampling from the following distribution

$$P_{remove\ shift}(x,y) = \sum_{j \in [L]} P_{LT}(y_j) P_{N_j,bal}(x|y_j) \tag{5}$$

is named as *removing shift sampling*. In fact, the obtained samples come from a "fake" dataset as the class distribution is predetermined instead counted from instances. We put the assumption and implementation details of removing shift sampling in Appendix B.1.

We perform ablation experiments on CIFAR-LT without or with removing shift sampling to investigate the effect of CCD shift on vanilla ERM (Vapnik, 1991) and some representative re-balancing methods: DRW Cao et al. (2019), CRT (Kang et al., 2019) and PC softmax (Menon et al., 2020; Hong et al., 2021). The main results and analysis are provided in the following subsections. We put experimental details and more results in Appendix B.

### 3.3 DISCOVERING THE UPPER BOUND OF LONG-TAIL RECOGNITION

Figure 1 and Figure 5 (in Appendix) quantitatively show how much CCD shift affects the performance in long-tailed recognition. The accuracy curves of all methods (vanilla ERM and re-balancing methods) exhibit significant gaps in performance with or without removing CCD shifts. And the gaps mainly happen for tail classes, indicating that the CCD estimations of tail classes are more unreliable. The results quantitatively verify that shift of CCD cannot be ignored and is the key factor to limit the performance in long-tail recognition.

Note that experiments with removing shift give an upper bound to the performance of these methods, as the empirical CCD from balanced datasets is not available when training on long-tailed datasets. The results suggest that it is promising and reasonable to improve generalization under long-tail distribution from the perspective of enhancing or against the unreliable empirical distribution.

### 3.4 ANALYZING RE-BALANCING METHODS FROM THE PERSPECTIVE OF CCD SHIFT

**Decoupling method avoids more severe CCD shift.** An interesting phenomenon in decoupling methods is imbalanced features are better than re-balanced features (Kang et al., 2019; Zhou et al., 2020). We explain that re-balancing in feature learning exacerbates unreliable empirical distributions because it assigns higher weights to unreliable empirical distributions of tail classes on expectations.

As a result, simply re-balancing harms feature representation since the influence of unreliable empirical distribution is even strengthened.

Table 1 quantifies the effect of CCD shift on re-balancing method CRT (Kang et al., 2019). Here, "feature re-balance" and "classifier adjust" indicate whether and what re-balancing method is applied in feature learning and classidier adjustment, respectively. It shows re-balanced[1] features are worse than uniformly learned imbalanced features. But this phenomenon does not appear after removing shift of CCD, as the re-balanced feature leads to better performance than the imbalanced feature. Hence, we have evidence that two-stage methods succeed partially because they avoid aggravating CCD shifts in feature learning. However, re-balancing in classifier adjustment may still suffer from CCD shift, leaving space to be improved, as do in (Zhang et al., 2021a; Wang et al., 2021).

**Sub-optimality of Fisher-consistent parameter in post-hoc adjustment.** Recall that post-hoc logit adjustment method adjusts logits to $f_\theta(x)_j + \tau log(P(y_j))$. The parameter $\tau = 1$ is proved Fisher-consistent, however, it is sub-optimal in experiments (Menon et al., 2020). We demystify this contradiction from the empirical study of CCD shift. Figure 2 shows logit adjustment gets best performance with $\tau$ much bigger than 1. We explain that rare instances from tail classes cannot provide enough information so the model regards them as "out of distribution" examples and gives low logits to their true class (Wang et al., 2021). Therefore more adjustment to tail classes, i.e. $\tau > 1$, gets better performance. The contradiction comes from the facts that logit adjustment assumes $P_{train}(x|y) = P_{test}(x|y)$ (Menon et al., 2020; Hong et al., 2021) but the shift of $P(x|y)$ really happens. As shown in Figure 2, with removing shift, $\tau = 1$ becomes optimal. Based on above analysis, after rectifying biased estimation on $P(y|x)$, further improvement of logit ad-

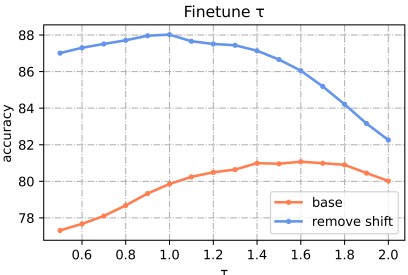

Figure 2: Accury on CIFAR10-LT with varying $\tau$ in post-hoc adjustment. After removing shift $\tau = 1$ gets optimal performance while it is sub-optimal with shift of $P(x|y)$.

justment lies in hardly recognizable samples due to CCD shift. Hence, it may be more reasonable to adjust logits by instance instead of by class to compensate for hard samples.

## 4 DISTRIBUTIONALLY ROBUST AUGMENTATION AGAINST CCD SHIFT

We have seen that shift of CCDs limits the performance of re-balancing methods. For unknown real distribution, minimization of the balanced risk is unobtainable even for Fisher-consistent methods (Gareth et al., 2013). To this end, we propose Distributionally Robust Augmentation (DRA) to learn a model more robust to the unreliable distribution estimation.

### 4.1 CONVERTING CLASS-AWARE ROBUSTNESS TO MIN-MAX OPTIMIZATION

With empirical distribution and a loss function, the balanced risk is approximated as:

$$R_{bal} = \frac{1}{L} \sum_{j \in [L]} E_{P_{N,LT}(x|y_j)}\{l_\theta(x, y)\} \tag{6}$$

As we have shown, empirical CCDs are unreliable to estimate real CCDs. Instead of minimizing the risk on empirical distribution, we utilize DRO (Kuhn et al., 2019) to **obtain a more robust model against unknown CCD shift**. In particular, we aim at minimizing the distributionally robust risk:

$$R_{DRO} = \frac{1}{L} \sum_{j \in [L]} \sup_{\hat{P}_j \in \mathcal{P}_j} E_{\hat{P}_j}\{l_\theta(x, y)\}, \tag{7}$$

---

[1]We choose DRW to get re-balanced features instead of RW (direct re-weighting) for RW gets the same result whether with removing shift sampling or not: it harms first-stage learning. Related explanation could be found in Appendix B.1.

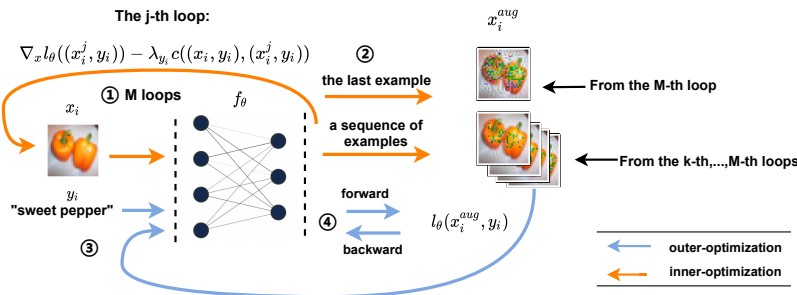

Figure 3: An illustration of DRA. Augmentation examples $x_i^{aug}$ generated from original data $x_i$ in $M$ inner loops (with two choices of the last example or a sequence of examples) are used as new training instances $(x_i^{aug}, y_i)$ to train a robust model against potential distribution shift. The order numbers indicate the process of DRA in an overall iteration.

in which $\hat{P}_j$ is a probability measure on $\mathcal{X} \times [L]$ and $\mathcal{P}_j = \{\hat{P}_j : W_c(\hat{P}_j, P_{N,LT}(x|y_j)) < r_j\}$. $W_c$ is Wasserstein distance induced by cost function $c((x_1, y_1), (x_2, y_2)) = \|x_1 - x_2\|_2 + \infty \cdot \mathbf{1}_{\{y_1 \neq y_2\}}$. Different from previous methods (Sinha et al., 2017; Shafieezadeh-Abadeh et al., 2019), **we set radii of uncertainty sets $\mathcal{P}_j$ dependent on classes to adapt to imbalanced instance numbers**, which implies different reliability by Proposition 1. Specifically, we set the radii decreased with the number of instances, i.e.$r_1 < \cdots < r_L$, to make the model robust enough without over-occupation of model capacity(Frogner et al., 2021; Zhang et al., 2020a).

Since $R_{DRO}$ in (7) defined on sets of distributions is intractable, we convert it to a min-max optimization problem on the training set, as presented in Theorem 4 (in the Appendix).

$$R_{DRO} = \inf_{\lambda_j \geq 0, j \in [L]} \frac{1}{L} \Big\{ \sum_{j \in [L]} \lambda_j r_j + E_{P_{N,LT}(x|y_j)} \Big\{ \sup_{z=(x_z, y_z) \in \mathcal{X} \times [L]} l_\theta(x_z, y_z) - \lambda_j \cdot c(z, (x, y)) \Big\} \Big\}. \tag{8}$$

Although the gradient of $\lambda_j$ can be computed by automatic differentiation, it brings another loop. Considering the radii of uncertainty sets need tuning as well, we relax $\lambda_j$ in (8) to simplify the optimization procedure without more hyperparameters. Then our objective changes from (8) to:

$$F(\theta) := \frac{1}{L} \sum_{j \in [L]} \sup_{\hat{P}_j} E_{\hat{P}_j} \{l(x, y)\} - \lambda_j \cdot W_c(\hat{P}_j, P_{N_{LT}, j}) \tag{9}$$

$$= \frac{1}{L} \sum_{j \in [L]} E_{P_{N,LT}(x|y_j)} \Big\{ \sup_{z=(x_z, y_z) \in \mathcal{X} \times [L]} l_\theta(x_z, y_z) - \lambda_j \cdot c(z, (x, y)) \Big\}. \tag{10}$$

The final objective (10) actually equals to the Lagrange penalty (9) for the original problem (7), as in the above equation. Besides, a generalization bound in Theorem 7 with mild conditions shows the risk on balanced label distribution with real CCD is partially bounded by the objective (10), meaning that our method provides a principled solution to the CCD shift.

## 4.2 LEARNING WITH A SEQUENCE OF AUGMENTED DATA

As illustrated in Figure 3, the learning process of DRA involves inner-optimization and outer-optimization, where the former generates augmentation examples from the original data, and the latter uses these augmented examples to train a robust model. In this way we minimize the converted robustness objective $F(\theta)$ by DRA.

More specifically, as the orange curve in Figure 3 indicates, we can directly solve the inner optimization $\sup_{z=(x_z, y_z) \in \mathcal{X} \times [L]} l_\theta(x_z, y_z) - \lambda_j \cdot c(z, (x, y))$ by gradient descent. Here $l_\theta(x_z, y_z)$ can be seen as mining harder examples for the current model, with $c(z, (x, y))$ constraining examples not too far away from the original instance and $\lambda_j$ determining the magnitude of the constraint. There are different ways to augment examples during inner optimization. One can rely on Algorithm 2 by just replacing $\lambda$ by our $\lambda_j$, which only uses the last point as augmentation example, annotated

as "the last example" in Figure 3. This strategy causes unstable optimization and bad performance when $\lambda_j$ are relatively small, as shown in Figure 4. To overcome these limitations, **we propose DRA Algorithm 1, which uses a sequence of examples, i.e. the last $s := M - k$ points, from the loops of inner-optimization**, annotated as "a squence of examples" in Figure 3. Empirically, our method achieves better and stable performance on small multipliers as in Figure 4.

Theoretically, we can prove a sequence of examples can make the optimization more stable as $s$ decreases the following convergency bound under common conditions of Sinha et al. (2017):

$$\frac{1}{T}\sum_{t\in[t]}E[\|\nabla_\theta F(\theta_t)\|_2^2] < \frac{1 - (C_1)^s}{s N_{batch}}C_2\epsilon + O(\sqrt{\frac{1}{N_{batch}T}}), \tag{11}$$

where $C_1 < 1, C_2$ are constants and $N_{batch}$ is the batch size. The bound is more formally presented in Theorem 5. More theoretical analysis (generalization and optimality guarantees) and implementation details of DRA could be seen in Appendix A, C.2.

## 5 EXPERIMENTS

We conduct experiments using DRA together with re-balancing methods e.g. delayed re-balancing, decoupling and logit adjustment on various long-tailed datasets. We also analyze the training cost of DRA and make comparison with other data augmentation methods. More complete experimental details and results are provided in Appendix C.

### 5.1 EXPERIMENTAL SETUP

**Comparative methods.** We compare our methods with (I) re-balancing methods including Decoupling (Kang et al., 2019), Logit Adjustment(Menon et al., 2020), PC softmax(Hong et al., 2021), DRS/DRW(Cao et al., 2019); (II) learned data augmentation including M2m(Kim et al., 2020), GIT(Zhou et al., 2022); (III) other related methods including LADE(Hong et al., 2021), DRO-LT(Samuel & Chechik, 2021), IB(Park et al., 2021), MFW(Ye et al., 2021), CDT(Ye et al., 2020).

**Datasets.** We conduct experiments on four real-world datasets including CIFAR10-LT, CIFAR100-LT, Tiny-ImageNet-LT Cao et al. (2019) and ImageNet-LT (Liu et al., 2019). We set the imbalance ratio of CIFAR-LT and Tiny-ImageNet-LT to 100. For all the benchmarks, we evaluate the performance by top-1 accuracy on the validation set as previous works (Hong et al., 2021; Cao et al., 2019; Menon et al., 2020).

### 5.2 MAIN RESULTS

Table 2 shows DRA improves all baseline re-balancing methods on three benchmarks in long-tailed recognition, e.g. DRA improves logit adjustment by 0.68% and 0.54% on CIFAR10-LT and CIFAR100-LT respectively. Note that LADE, also based on logit adjustment, improves by 0.22% and 0.22% respectively. Our DRA even marginally outperforms GIT and M2m which use learned augmentation but need additional model (e.g. a GAN (Goodfellow et al., 2014)) and training process apart from end-to-end training. We conduct a more detailed comparison with these data augmentation methods in ablation study.

Table 7 (in Appendix) shows that DRA is also effective on the large-scale dataset ImageNet-LT. Similarly, DRA shows considerable improvement on re-balancing methods, building stronger baselines for long-tailed recognition. Moreover, the improvement gained from DRA is not only from few-shot but also from many-shot and medium-shot, which verifies distributionally robustness could enhance the generalization of head classes as well.

### 5.3 ABLATION STUDY

**Class-aware uncertainty set.** As explained in Appendix C.2, we set class-aware multiplier as $\lambda_j = normalize\{N_j^\beta\} * S$ with hyperparameters $\beta$ and $S$. Through $\beta$ we assign different robustness to classes. $\beta = 0$ gives every class the same robustness and DRA reduces to WRM. With different values of $\beta$ and $S$, we can reveal the effects of class-aware radius and validate our hypothesis.

Table 2: Comparison of top-1 accuracy (%) on different benchmarks. - denotes results not reported, † means results from original papers, ‡ means reproduced results from official codes, * means requiring more training epochs. Best results are in bold and small red font denotes performance gain.

| Method | CIFAR10-LT | CIFAR100-LT | Tiny-ImageNet-LT |
|---|---|---|---|
| ERM | 73.24 | 40.28 | 34.20 |
| CE-DRS | 77.21 | 44.16 | 36.02 |
| LDAM-DRW | 78.67 | 45.05 | 37.43 |
| LDAM-DRS | 78.71 | 45.12 | 37.66 |
| PC softmax | 79.58 | 44.34 | - |
| Logit Adjustment | 79.65 | 45.17 | - |
| LDAM-M2m† | 79.1 | 43.4 | - |
| CE-DRS-GIT | 77.35‡ | 43.02‡ | 17.43† |
| LDAM-DRS-GIT | 79.36‡ | 43.65‡ | 21.99† |
| MFW*† | 78.5 | 44.7 | 35.4 |
| CDT† | 79.4 | 44.3 | 37.9 |
| IB‡ | 78.86 | 43.18 | **40.40** |
| LADE‡ | 79.87 | 45.39 | - |
| DRO-LT*‡ | 79.1 | 44.2 | - |
| **CE-DRA** | 75.08+1.84 | 41.37+1.09 | 35.11+0.91 |
| **CE-DRS-DRA** | 78.89+1.68 | 44.87+0.71 | 36.18+0.12 |
| **LDAM-DRS-DRA** | 79.75+1.04 | 45.54+0.42 | 39.07+1.41 |
| **PC softmax-DRA** | **80.39**+0.81 | 44.84+0.50 | - |
| **Logit Adjustment-DRA** | 80.33+0.68 | **45.71**+0.54 | - |

**The number of augmentation examples.** We perform experiments to verify the effect of using a sequence of points as augmentation examples in DRA to calculate outer gradient. Specifically, we set $k$ in DRA to $M/2$ or $M - 1$ and obtain comparative results.

Figure 4 shows the effectiveness of DRA in two innovations of class-aware uncertainty set and the number of augmentation examples. As the bound (11) indicates, a sequence of augmentation examples is more stable than the last point. Class-aware radius exhibits significant superiority while class-consistent radius (in WRM) only slightly improves the performance.

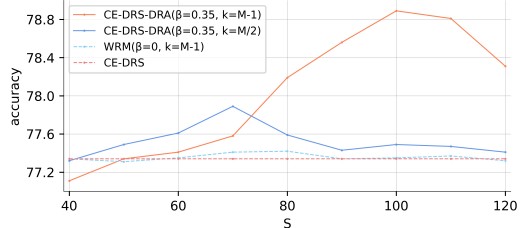

Figure 4: Accuracy on CIFAR10-LT with respect to different choices of hyperparameters $S$, $\beta$ and $k$ in inner optimization

**More ablation study.** We put more ablation study in Appendix C.4, including comparison with augmentation methods, training cost, confidence calibration, visualization and discussion.

## 6    CONCLUSION

In this paper, we discover empirical evidence that unreliable distribution estimation in long-tailed recognition, exhibiting as shift of $P(x|y)$, is a key factor to limit performance even with re-balancing methods. It suggests that only regarding long-tailed learning as label distribution shift problem is not reasonable enough. By our proposed DRA, we show training a model robust to potential distribution shift is effective to improve long-tailed recognition with theoretical and empirical guarantees. Space to improve e.g. more precise uncertain set for potential shift is considered possible. Overall, to some degree our work sheds light on the bottleneck of long-tailed recognition and encourages further solutions under other settings from the perspective of unreliable empirical estimation.

## 7 REPRODUCIBILITY STATEMENT

For our empirical study experiments, Appendix B.1 describes the implementation details such as the probability setting of removing shift sampler, model architecture, and hyperparameters. In our main comparative experiments, for methods used jointly with DRA, we implement them based on official codes and keep default hyperparameters for their modules. For other methods, we use results reproduced from publicly available official codes or numbers reported in the original papers. Appendix C provides detailed experiment settings such as model architecture, optimization hyperparameters, and the implementation of DRA.The pseudocode of DRA is in Algorithm 1. Besides, we put all proofs of propositions and theorems presented in this paper in Appendix A.

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

# A    PROOFS AND MORE THEORETICAL ANALYSIS

**Lemma 2 (Kantorovich-Rubinstein Theorem)** *(Villani, 2009, 5.16)  For Wassernstein distance $W_c$ between two distributions $Q$ and $Q'$, $Lip(f) = \sup_{x_1,x_2} \frac{|f(x_1)-f(x_2)|}{\|x_1-x_2\|}$, it admits*

$$W_c(Q, Q') = \sup_{Lip(\phi)<1} \int_{\mathbb{R}^m} \phi(\xi)Q(d\xi) - \int_{\mathbb{R}^m} \phi(\xi)Q'(d\xi).$$

**Lemma 3 (Measure concentration)** *(Fournier & Guillin, 2015, Theorem 2)  For a probability distribution $P$ on $\mathbb{R}^m$ and $P_N$ is the empirical distribution of $N$ instances i.i.d sampled from $P$, $A = E_{P(x|y)}\{\|x\|^{\alpha}\} < \infty$ for some $\alpha > 0$, then $\exists c_1, c_2$, depending on $A$, $\alpha$, $P(W_c(P, P_N) < \epsilon) > 1 - \eta$ holds, for*

$$\epsilon(\eta) = \begin{cases} (\frac{log(c_1/\eta)}{c_2 N})^{1/m} \text{ if } N \geq \frac{log(c_1/\eta)}{c_2} \\ (\frac{log(c_1/\eta)}{c_2 N})^{1/\alpha} \text{ if } N < \frac{log(c_1/\eta)}{c_2} \end{cases}$$

## A.1    PROOF OF PROPOSITON 1

$$\left| E_{P(x|y)}\{l(x,y)\} - E_{P_N(x|y)}\{l(x,y)\} \right|$$
$$= L \cdot \left| E_{P(x|y)}\{l(x,y)/L\} - E_{P_N(x|y)}\{l(x,y)/L\} \right|$$
$$\leq L \cdot W_c(P(x|y), P_N(x|y))$$

The last equality is from Lemma 2. And directly substituting $\eta$ in Lemma 3 by $\frac{c_1}{e^{(\frac{t}{L})^{min\{m,a\}}c_2 N}}$ gets the result.

## A.2 PROOF OF THEOREM 4

**Theorem 4** *For a continuous loss $l : \Theta \times (\mathcal{X} \times [L]) \to \mathbb{R}$, we have*

$$R_{DRO} = \inf_{\lambda_j \geq 0, j \in [L]} \frac{1}{L} \{ \lambda_j r_j + \sum_{j \in [L]} E_{P_{N,LT}(x|y_j)} \{ \sup_{z=(x_z, y_z) \in \mathcal{X} \times [L]} l_\theta(x_z, y_z) - \lambda_j \cdot c(z, (x, y)) \} \}.$$

(12)

*Considering Lagrange penalty with nonnegative multipliers $\{\lambda_j\}_{j \in [L]}$, we have the reformulation:*

$$F(\theta) := \frac{1}{L} \sum_{j \in [L]} \sup_{\hat{P}_j} E_{\hat{P}_j} \{ l(x, y) \} - \lambda_j \cdot W_c(\hat{P}_j, P_{N_{LT}, j})$$

(13)

$$= \frac{1}{L} \sum_{j \in [L]} E_{P_{N,LT}(x|y_j)} \{ \sup_{z=(x_z, y_z) \in \mathcal{X} \times [L]} l_\theta(x_z, y_z) - \lambda_j \cdot c(z, (x, y)) \}.$$

(14)

Our proof generalizes the proof in Sinha et al. (2017), which is a special case $\lambda_j \equiv C$ of ours.

Using the duality firstly, we have

$$R_{DRO} = \frac{1}{L} \sum_{j \in [L]} \sup_{\hat{P}_j \in \mathcal{P}_j} E_{\hat{P}_j}[l_\theta(x, y)]$$

$$= \sup_{\hat{P}_j \in \mathcal{P}_j} \inf_{\lambda_j \geq 0} \frac{1}{L} \sum_{j \in [L]} \{ E_{\hat{P}_j}[l_\theta(x, y)] - \lambda_j W_c(\hat{P}_j, P_{N,LT}(x|y_j)) + \lambda_j r_j \}$$

$$= \inf_{\lambda_j \geq 0} \{ \lambda_j r_j + \sup_{\hat{P}_j \in \mathcal{P}_j} \frac{1}{L} \sum_{j \in [L]} \{ E_{\hat{P}_j}[l_\theta(x, y)] - \lambda_j W_c(\hat{P}_j, P_{N,LT}(x|y_j)) \} \}.$$

The last equality is based on strong duality which will be explained later. All we need to prove now is

$$\sup_{\hat{P}_j \in \mathcal{P}_j} \frac{1}{L} \sum_{j \in [L]} \{ E_{\hat{P}_j}[l_\theta(x, y)] - \lambda_j W_c(\hat{P}_j, P_{N,LT}(x|y_j)) \}$$

$$= \frac{1}{L} \sum_{j \in [L]} E_{P_{N,LT}(x|y_j)} [ \sup_{z=(x_z, y_z) \in \mathcal{X} \times [L]} l_\theta(x_z, y_z) - \lambda_j \cdot c(z, (x, y)) ].$$

Actually, we have

$$\sup_{\hat{P}_j \in \mathcal{P}_j} \frac{1}{L} \sum_{j \in [L]} \{ E_{\hat{P}_j}[l_\theta(x, y)] - \lambda_j W_c(\hat{P}_j, P_{N,LT}(x|y_j)) \}$$

$$= \sup_{\hat{P}_j \in \mathcal{P}_j} \frac{1}{L} \sum_{j \in [L]} \int l_\theta(\xi, y) \hat{P}_j(d\xi) - \lambda_j \inf_{\mathcal{M} \in \pi(\hat{P}_j, P_{N,LT}(x|y_j))} \int c(\xi_1, \xi_2) \, \mathcal{M}(d\xi_1, d\xi_2)$$

$$= \sup_{\hat{P}_j \in \mathcal{P}_j, \mathcal{M} \in \pi(\hat{P}_j, P_{N,LT}(x|y_j))} \frac{1}{L} \sum_{j \in [L]} \int l_\theta(\xi, y) - \lambda_j c((\xi, y), \xi_2) \, \mathcal{M}((d\xi, y), d\xi_2)$$

$$\leq \sup_{\hat{P}_j \in \mathcal{P}_j, \mathcal{M} \in \pi(\hat{P}_j, P_{N,LT}(x|y_j))} \frac{1}{L} \sum_{j \in [L]} \int \sup_{z=(x_z, y_z) \in \mathcal{X} \times [L]} l_\theta(x_z, y_z) - \lambda_j c((x_z, y_z), \xi_2) \, \mathcal{M}(d\xi_1, d\xi_2)$$

$$= \frac{1}{L} \sum_{j \in [L]} E_{P_{N,LT}(x|y_j)} [ \sup_{z=(x_z, y_z) \in \mathcal{X} \times [L]} l_\theta(x_z, y_z) - \lambda_j \cdot c(z, (x, y)) ].$$

The first equality is from the definition of Wasserstein distance and the last equality is from $\sup_{z=(x_z, y_z) \in \mathcal{X} \times [L]} l_\theta(x_z, y_z) - \lambda_j \cdot c(z, \xi_2)$ is independent of $\xi_1$. We now explain that the inequality is actually an equality. For every $i$, we can find $z_i^* = (x_z^*, y_i)$ for every $\epsilon > 0, k \in N$

satisfying

$$l_\theta(x_z^*, y_i) - \lambda_j \cdot c(z_i^*, (x_i, y_i)) \geq \sup_{z=(x_z,y_z)\in\mathcal{X}\times[L]} l_\theta(x_z, y_z) - \lambda_j \cdot c(z, (x_i, y_i)) - \epsilon$$

$$\text{if} \quad \sup_{z=(x_z,y_z)\in\mathcal{X}\times[L]} l_\theta(x_z, y_z) - \lambda_j \cdot c(z, (x_i, y_i)) < \infty$$

$$l_\theta(x_z^*, y_i) - \lambda_j \cdot c(z_i^*, (x_i, y_i)) \geq k$$

$$\text{if} \quad \sup_{z=(x_z,y_z)\in\mathcal{X}\times[L]} l_\theta(x_z, y_z) - \lambda_j \cdot c(z, (x_i, y_i)) = \infty$$

Let $\mathcal{M}(\xi_1, \xi_2) = \frac{1}{N} \sum_{i\in[N]} \delta(z_i^*, (x_i, y_i))$, and for arbitrariness of $\epsilon$ or $k$, the equality is established.

Finally, we prove the strong duality. Let $\hat{P}_j = P_{N_{LT,j}}(x|y)$, and $W_c(\hat{P}_j, P_{N_{LT,j}}(x|y)) = 0$. Therefore, $\{0\cdots0\} \in \mathbb{R}^L$ is the inner point of the set $\{b \in \mathbb{R}^L | b_j = \frac{1}{N_j} \sum_{y_i=j} \int \|\xi - x_i\|_2 Q(d\xi), Q$ is a probability measure on $\mathcal{X}\}$, satisfying the slater's condition of a standard conic programming duality result (Shapiro, 2001, 3.4), thus the strong duality is established.

### A.3 PROOF OF THEOREM 5

**Theorem 5** *Under similar conditions to WRM (Assumption 1, 2,3), let $T$ be the number of iterations in outer optimization, $\Delta_F = F(\theta_0) - \inf_{n\in[T]} F(\theta_n)$, $N_{batch}$ is the batchsize, with stepsize of outer optimization $\alpha = \min\{\frac{1}{2L_\Phi}, \sqrt{\frac{N_{batch}\Delta_F}{L_\Phi T\sigma^2}}\}$ and inner stepsize $\alpha_{inner} = \frac{1}{\lambda - L_{zz}}, L_\phi = L_{\theta\theta} + \frac{L_{z\theta}L_{\theta z}}{\lambda - L_{zz}}$ with $L_{\theta z}, L_{\theta\theta}, L_{z\theta}, L_{zz}, \sigma$ depending on $l$. For Algorithm 1, let $s = M - k$ is the number of examples, $\lambda = min_{j\in[L]}\lambda_j, G = \max\{L_{\theta z}, L_{\theta\theta}, L_{z\theta}, L_{zz}\}$, then for $\{\theta_j\}_{j=0}^T$ in the outer optimization, we have*

$$\frac{1}{T} \sum_{t\in[t]} E[\|\nabla_\theta F(\theta_t)\|_2^2] < \frac{1}{sN_{batch}} \frac{1 - (1 - \frac{\lambda - L_{zz}}{G})^s}{\frac{\lambda - L_{zz}}{G}} \frac{6L_{\theta z}^2}{\lambda - L_{zz}} \epsilon + 4\sigma\sqrt{\frac{1}{N_{batch}} \frac{L_\Phi \Delta_F}{T}}. \quad (15)$$

Actually, in the inequality (11), considering that $\lambda_j$ is small but concavity is established ($\lambda_j \geq L_{zz}$), we assume $\frac{\lambda_j - L_{zz}}{G} < 1$, then the first term of the bound reduces with the increase of the number of examples $s$ thus the overall optimization becomes more stable. DRA provides a principled solution to CCD shift as long as real CCDs are included by the candidates we consider.

With similar mild assumptions as WRM (Sinha et al., 2017) below, one can prove that our Distributionally Robust Augmentation Algorithm 1 enjoys a guarantee of convergence. It finds a $\epsilon$-stationary point of model parameters $\theta$ with $O(\epsilon^2)$ iterations (as fast as SGD (Amari, 1993)).

**Assumption 1** *The loss $l : \theta \times \mathcal{Z}(\equiv \mathcal{X} \times [L]) \to \mathbb{R}$ satisfies the smoothness conditions:*

$$\left\|\nabla_\theta l(\theta, z) - \nabla_\theta l(\theta', z)\right\|_* \leq L_{\theta\theta} \|\theta - \theta^*\|, \left\|\nabla_z l(\theta, z) - \nabla_z l(\theta, z')\right\|_* \leq L_{zz} \|z - z^*\|,$$

$$\left\|\nabla_\theta l(\theta, z) - \nabla_\theta l(\theta, z')\right\|_* \leq L_{\theta z} \|z - z^*\|, \left\|\nabla_z l(\theta, z) - \nabla_z l(\theta', z)\right\|_* \leq L_{z\theta} \|\theta - \theta^*\|.$$

$\|\cdot\|_*$ *is the dual norm of the norm $\|\cdot\|$ (as our settings, $\|\cdot\|_2$).*

We also borrow a lemma from WRM as follows:

**Lemma 6** *(Sinha et al., 2017, Lemma 1) Let $f : \theta \times \mathcal{Z} \to \mathbb{R}$ be differentiable and $\eta$-strongly concave in $z$, define $\overline{f}(\theta) = \sup_{z\in\mathcal{Z}} f(\theta, z)$. Let $g_\theta(\theta, z) = \nabla_\theta f(\theta, z)$ and $g_z(\theta, z) = \nabla_z f(\theta, z)$ and $f$ satisfies Assumption 1 with replacing $l$ with $f$. Then $\overline{f}$ is differentiable and $\nabla_\theta \overline{f} = \nabla_\theta f(\theta, z^*(\theta))$ where $z^*(\theta) = \sup_{z\in\mathcal{Z}} f(\theta, z)$. And we have*

$$\|z^*(\theta_1) - z^*(\theta_2)\| \leq \frac{L_{z\theta}}{\eta} \|\theta_1 - \theta_2\|,$$

*and*

$$\left\| \nabla \overline{f}(\theta) - \nabla \overline{f}(\theta^*) \right\|_* \le (L_{\theta\theta} + \frac{L_{z\theta}L_{\theta z}}{\eta}) \left\| \theta_1 - \theta_2 \right\|.$$

Under Assumption 1, the primal of inner optimization $l(\theta, z) - \lambda_j c(z, z_0)$ is $(\lambda_j - L_{zz})$-strongly concave, so that $z_{j,0}^* = \arg\sup_z l(\theta, z) - \lambda_j c(z, z_0)$ is well-defined and satisfies the condition of Lemma 6. Thus, we have $\nabla_\theta \sup_{z \in \mathcal{Z}} l(\theta, z) - \lambda_j c(z, z_0) = \nabla_\theta l_\theta(z_{j,0}^*)$ and $\nabla_\theta F(\theta)$ is $\frac{L_{z\theta}L_{\theta z}}{\lambda_j - L_{zz}}$-Lipschitz. Then we make an assumption on the variance of the gradient in the outer-optimization, which is a common condition when analyzing the convergence of SGD.

**Assumption 2** *For any sampled training set $\{(x_i, y_i)\}_{i \in [N]}$, it holds*

$$E \left\| \nabla_\theta l_\theta(z_{y_i, i}^*) - \nabla_\theta F(\theta) \right\|^2 \le \sigma^2,$$

*where $z_{y_i, i}^* = \arg\sup_z l_\theta(z) - \lambda_j c(z, (x_i, y_i))$.*

With the preparation above, we begin to prove Theorem 5. Let $h_j(\theta, z; (x_i, j)) := l(\theta, z) - \lambda_j c(z, (x_i, j))$. In Algorithm 1, the gradient we use to update is actually

$$g_t = \frac{1}{N_{batch}} \sum_{i \in [N_{batch}]} \frac{1}{s} \sum_{r \in [s]} \nabla_\theta h_{y_i}(\theta_t, (x_i^{r+k}, y_i); (x_i, y_i)).$$

Since the gradient of $F(\theta)$ is $L_\phi$-smooth,

$$F(\theta_{t+1}) < F(\theta_t) + \langle \nabla F(\theta_t), \theta_{t+1} - \theta_t \rangle + \frac{L_\phi}{2} \left\| \theta_{t+1} - \theta_t \right\|_2^2$$

$$= F(\theta_t) - \alpha(1 - \frac{L_\phi \alpha}{2}) \left\| \nabla F(\theta_t) \right\|_2^2 + \alpha(1 + \frac{L_\phi \alpha}{2}) \langle \nabla F(\theta_t), F(\theta_t) - g_t \rangle$$

$$+ \frac{L_\phi \alpha^2}{2} \left\| F(\theta_t) - g_t \right\|_2^2.$$

We need to estimate $F(\theta_t) - g_t$ in the equation above. Let

$$g_t^* = \frac{1}{N_{batch}} \nabla_\theta \sum_{i \in [N_{batch}]} l(\theta_t, z_{y_i, i}^*(\theta_t)).$$

With the help of $g_t^*$, we can estimate $F(\theta_t) - g_t$ more easily. The difference between $g_t$ and $g_t^*$ comes from we use a series of points to compute the gradient to $\theta$ and the inner optimization cannot obtain an optimal solution precisely. The difference between $\nabla F(\theta_t)$ and $g_t^*$ is from just sampling a batch from all training instances.

We compute the difference between $g_t$ and $g_t^*$ as $\delta_t = g_t - g_t^*$, then we have

$$\|\delta_t\|_2^2 = \left\| \frac{1}{N_{batch}} \sum_{i \in N_{batch}} \{ \frac{1}{s} \sum_{r \in [s]} \nabla_\theta l(\theta_t, (x_i^{r+k}, y_i)) - \nabla_\theta l(\theta_t, z_{y_i, i}^*) \} \right\|_2^2$$

$$\le (\frac{1}{N_{batch}})^2 \sum_{i \in N_{batch}} (\frac{1}{s})^2 \sum_{r \in [s]} l_{\theta z}^2 \left\| (x_i^{r+k}, y_i) - z_{y_i, i}^* \right\|_2^2.$$

**Assumption 3** *The inner-optimization just reaches the linear convergence rate (Nesterov, 1998) of gradient descent in strongly-convex optimization i.e.* $\left\| (x_i^{k+s}, y_i) - z_{y_i, i}^* \right\|_2^2 \le (1 - \frac{\lambda - L_{zz}}{G})^{k+s} \left\| (x_i^0, y_i) - z_{y_i, i}^* \right\|_2^2 \le \epsilon.$

We make this assumption under the motivation that the last example is enough when the convergence is much faster than the convergence bound that the strongly-convex condition gives. However when the convergence of inner-optimization is slow i.e. just reaches the convergence bound, using a sequence of examples can gain benefits.

We have $\left\|(x_i^{k+s}, y_i) - z_{y_i,i}^*)\right\|_2^2 \leq \frac{1}{\lambda_{y_i} - L_{zz}}\epsilon$. With another assumption that inner-optimization reaches the linear convergence rate of gradient descent on strongly concave optimization (Nesterov, 1998), as the inner optimization is at least $\lambda - L_{zz}$-strongly concave, we have

$$\left\|(x_i^{k+r}, y_i) - z_{y_i,i}^*)\right\|_2^2 \leq (1 - \frac{\lambda - L_{zz}}{G})^{s-r}\frac{1}{\lambda_{y_i} - L_{zz}}\epsilon.$$

As a result, $\|\delta_t\|_2^2 \leq \frac{1}{sN_{batch}}\frac{1-(1-\frac{\lambda-L_{zz}}{G})^s}{\frac{\lambda-L_{zz}}{G}}\frac{4L_{\theta z}^2}{\lambda-L_{zz}}\epsilon.$

Substituing $g_t$ with $\delta_t$, we have

$$F(\theta_{t+1}) < F(\theta_t) - \frac{\alpha}{2}\|\nabla F(\theta_t)\|_2^2 + \frac{\alpha}{2}(1 - \frac{1}{2}L_\phi\alpha)\|\delta_t\|_2^2$$
$$+ \alpha(1 - L_\phi\alpha)\langle\nabla F(\theta_t), \nabla F(\theta_t) - g_t^*\rangle + L_\phi\alpha^2(\|\delta_t\|_2^2 + \|\nabla F(\theta_t) - g_t^*\|_2^2).$$

For Lemma 6, $g_t^*$ is unbiased estimation to $\nabla F(\theta_t)$ so $E[g_t^* - \nabla F(\theta_t)|\theta_t] = 0$. Using Assumption 2 to control the invariance of the estimation and taking expectations of the above formula, we have

$$E[F(\theta_{t+1}) - F(\theta_t)] \geq -\frac{\alpha}{2}E[\|\nabla F(\theta_t)\|_2^2]$$

$$+\frac{1}{N_{batch}}L_\phi\alpha^2\sigma^2 + \frac{3}{4}\alpha\frac{1}{sN_{batch}}\frac{1 - (1 - \frac{\lambda - L_{zz}}{G})^s}{\frac{\lambda - L_{zz}}{G}}\frac{4L_{\theta z}^2}{\lambda - L_{zz}}\epsilon,$$

where we use $\alpha < \frac{1}{2L_\phi}$ to get $\frac{3}{2}L_\phi\alpha^2 \leq \frac{3}{4}\alpha$.

Summing by $t$, we get

$$\frac{1}{T}\sum_{t=1}^T E[\|\nabla F(\theta_t)\|_2^2] \leq 2\frac{\Delta F}{\alpha T} + \frac{2}{N_{batch}}L_\phi\alpha\sigma^2 + \frac{6}{sN_{batch}}\frac{1 - (1 - \frac{\lambda - L_{zz}}{G})^s}{\frac{\lambda - L_{zz}}{G}}\frac{4L_{\theta z}^2}{\lambda - L_{zz}}\epsilon.$$

Substituting with the stepsize $\alpha$, the conclusion is obtained.

### A.4 GENELIZATION BOUND AND OPTIMALITY OF AUGMENTATION EXAMPLES

In addition, we give a bound on the estimated balanced risk on real distribution and the optimality of examples generated by DRA under mild conditions.

**Theorem 7 (Genelization bound)** *Assuming $l : \Theta \times (\mathcal{X} \times [L]) \to \mathbb{R}$ is continious and for every class $j$, let the real class-conditional distribution $P(x|y_j)$ satisfies $\exists \alpha > 0, A = E_{P(x|y_j)}\{\|\xi\|^\alpha\} < \infty$, then $\exists c_1, c_2$ only depending on $A, \alpha$, for any multipier $\{\lambda_j\}_{j\in[L]}$, with probability $1 - \eta$ it holds*

$$R_{bal,l} \leq \sum_{j\in[L]}\frac{1}{L}\{\lambda_j \max\{(\frac{log(c_1/\eta)}{c_2 N_j})^{1/m}, (\frac{log(c_1/\eta)}{c_2 N_j})^{1/\alpha}\}$$
$$+ E_{P_{N,LT}(x|y_j)}[\sup_{z=(x_z,y_z)\in\mathcal{X}\times[L]} l(x_z, y_z) - \lambda_j \cdot c(z, (x,y))]\},$$

*in which $R_{bal,l} := \sum_{j\in[L]}\frac{1}{L}E_{P(x|y_j)}\{l(x,y)\}$ is the balanced risk on test distribution $P_{test} = \frac{1}{L}\sum_{j\in[L]}P(x|y_j)$ estimated by loss function $l(x,y)$.*

**Remark 1** *The above bound shows a new viewpoint beyond existing theoritical results of long-tailed recogniton in the community. We make some remarks for the bound as follows:*

1. *It is more flexible and tight than the results of prior DRO works (Sinha et al., 2017; Kuhn et al., 2019). It adjusts class-wise robustness by $\lambda_j$ for each class $j$, while prior work only admits a special case of $\lambda_j = C$, for all $j \in [L]$.*

2. *It is closer to practical training compared to the Fisher-consistency result by logit adjustment (Menon et al., 2020). Fisher-consistency theory from Theorem 1 in Menon et al. (2020) only states that a Bayes-optimal classifier would be obtained by the minimization of logit adjustment loss under a balanced label distribution with real CCDs.*

    *3. It is no longer dependent on the capacity of the hypothesis class, which is extremely large for modern neural networks, while the bound in Theorem 1 of Cao et al. (2019) is.*

This bound shows with a high probability the balanced risk on real distribution could be bounded by our objective (10) plus a constant depending on the multiplier $\lambda_j$ and the number of instances from each class, which serves as a generalization gap, decreasing with the number of instances $N_j$. It indicates $\lambda_j$ controls a trade-off between robustness and performance. If we do not enforce any robustness i.e. the multipiers $\lambda_j$ are extremely large, DRA actually equals to ERM. In this situation, large $\lambda_j$ leads to considerable generalization gap, especially for the tail classes. Hence, **we conclude that models robust to CCD shift benefit long-tailed recognition.** Conversely, making $\lambda_j$ extremely small near zero, it admits a trivial bound meaning the model refuses to make predictions, which is well-known as "over-pessimism" in DRO (Frogner et al., 2021).

**Proposition 8 (Optimality of augmentation examples)** *Assuming $l : \Theta \times (\mathcal{X} \times [L]) \to \mathbb{R}$ is continuous. $T_j((x,y)) := arg\max_{z=(x_z,y_z)\in\mathcal{X}\times[L]} l(x_z, y_z) - \lambda_j \cdot c(z,(x,y))$ is the optimal solution of inner optimization, $\mathcal{M}$ is the set of probability measure on $\mathcal{X} \times [L]$ and $P^*_{N,LT}(x|y_j) = \frac{1}{N_{LT,j}} \sum_{\{i:y_i=y_j\}} \delta(T_j((x_i,y_i)))$ is the empirical distribution consisting of them. Then for any multipier $\{\lambda_j\}_{j\in[L]}$, we have*

$$P^*_{N,LT}(x|y_j) = \arg\max_{\hat{P}_j\in\mathcal{M}} E_{\hat{P}_j}[l(x,y)] - \lambda_j W_c(\hat{P}_j, P_{N,LT}(x|y_j)),$$

*which indicates the empirical distribution consisting of ideal optimal solutions of inner optimization of a class is the optimal perturbed distribution of the Lagrange penalty problem.*

### A.5 PROOF OF THEOREM 7

Using Theorem 4, for any multiplier $\{\lambda_j\}_{j\in[L]}$, we have

$$\frac{1}{L} \sum_{j\in[L]} \sup_{\hat{P}_j\in\mathcal{P}_j} E_{\hat{P}_j}\{l(x,y)\} < \frac{1}{L} \sum_{j\in[L]} \lambda_j r_j$$
$$+ E_{P_{N,LT}(x|y_j)}[\sup_{z=(x_z,y_z)\in\mathcal{X}\times[L]} l(x_z,y_z) - \lambda_j \cdot c(z,(x,y))],$$

in which $r_j$ is the radius of $\mathcal{P}_j$.

And with our assumption, $P(x|y_j)$ satisfies conditions of Lemma 3. So let $r_j = \max\{(\frac{log(c_1/\eta)}{c_2 N_j})^{1/m}, (\frac{log(c_1/\eta)}{c_2 N_j})^{1/\alpha}\}$, we have $P(x|y_j) \in \mathcal{P}_j$ with the probability of $1-\eta$. Thus, it establishes

$$\sum_{j\in[L]} \frac{1}{L} E_{P(x|y)}\{l(x,y)\} < \frac{1}{L} \sum_{j\in[L]} \sup_{\hat{P}_j\in\mathcal{P}_j} E_{\hat{P}_j}\{l(x,y)\}$$

and we get the conclusion.

### A.6 PROOF OF PROPOSITION 8

$$\sup_{\hat{P}_j\in\mathcal{M}} E_{\hat{P}_j}[l(x,y)] - \lambda_j W_c(\hat{P}_j, P_{N,LT}(x|y_j))$$
$$= E_{P_{N_{LT,j}}(x|y)}[\sup_{z=(x_z,y_z)\in\mathcal{X}\times[L]} l(x_z,y_z) - \lambda_j \cdot c(z,(x,y))]$$
$$= E_{P^*_{N,LT}(x|y_j)}[l(x,y)] - \lambda_j \sum_{\{i:y_i=y_j\}} c(T_j((x_i,y_i)),(x_i,y_i))$$
$$\leq E_{P^*_{N,LT}(x|y_j)}[l(x,y)] - \lambda_j W_c(P^*_{N,LT}(x|y_j), P_{N,LT}(x|y_j))$$

The first equality is from Theorem 4 and the second is from the definition of $P^*_{N,LT}(x|y_j)$. The inequality is from the definition of Wasserstein distance. For the inequality in the opposite direction is trivial, we establish the conclusion.

# B  DETAILED EMPIRICAL STUDY ON CCD SHIFT

## B.1  EXPERIMENTS SETTING

**Assumption of removing shift.** To conduct experimental comparisons on CCD, we propose removing shift sampling in Section 3.2. An ideal situation for comparision on CCD is to obtain real CCD $P(x|y)$ and keep imbalanced label distribution $P(y)$. In fact, due to no access to the real $P(x|y)$, we make **the empirical CCD $\mathrm{P}_{N,bal}(x|y)$ serve as an oracle to substitute real CCD $\mathrm{P}(x|y)$**. At the same time, we keep imbalance by using the same label distribution $P_{LT}(y)$ as long-tailed datasets so that the label distribution is imbalanced in every batch and so it is in the whole training. In this way, more unique instances are seen by the model but the number of samples from each class, i.e. label distribution, keeps unchanged in all batches during training. So under our assumption, we make a fair ablation study on CCD with proper control of other factors. It is possible that $P_{N,bal}(x|y)$ cannot estimate real CCD well, but as long as the oracle brings improvement in performance, we can blame CCD shift correctly. The counterexample appears in extreme cases when the oracle does not make improvement. In that case, we cannot tell whether the CCD shift does not appear thus should not be blamed or our oracle is too weak to identify the CCD shift.

**Implementation of removing shift sampling.** We do removing shift sampling on original balanced dataset CIFAR10/100 (Krizhevsky, 2009). We use a dataloader with a class-imbalanced sampler which samples with the same class probability $P_{LT}(y)$ as the long-tailed dataset and samples uniformly within each class. The accumulated probabilities of instances from each class equals the predetermined class frequency and every instance within a class has the same sampling probability. To this end, we implement the sampler as a multinomial distribution sampler in the same way as that used to generate CIFAR-LT in Cao et al. (2019). Different from the generation of long-tailed dataset, we sample from the whole balanced dataset and get class-imbalanced instances in every batch. In addition, We keep the same amount of data as the long-tailed dataset while only change the class-conditional distribution via removing shift in the ablation experiment.

**More implementation details.** We train ResNet (He et al., 2016) with batch size 256, optimized by SGD with momentum 0.9 and weight decay $2 \times 10^{-4}$ with warm-up scheduler. The learning rate decays by a factor of 0.01 at epochs 160 and 180 with an initial rate 0.2. We evaluate Top-1 accuracy on the original validation set of the datasets, following the common protocol in long-tailed recognition (Cao et al., 2019).

**Methods in ablation study.** We select ERM as baseline and DRW, PC softmax (equals to post-hoc logit adjustment) and CRT as representative re-balancing methods.

1. **ERM.** ERM means empirical risk minimization (Vapnik, 1991). We train a model on long-tailed datasets with cross-entropy loss without any strategy.

2. **Balance.** For easy reference, we also train a model under our setting on the original CIFAR10/100.

3. **DRW.** We implement DRW (Cao et al., 2019) following the original paper. We train in a regular way in the early phase and re-weight the loss by the inverse of the predetermined class frequencies only in the last phase during training. We typically re-weight the loss function starting at 160 epochs.

4. **RW.** RW means directly re-weighting the loss by the inverse of the predetermined class frequencies. We re-weight the loss in the whole training phase. In ablation experiments on decoupling methods, we utilize DRW and RW to obtain re-balanced features and ERM to obtain imbalanced features in the first stage of training.

5. **PC softmax.** We implement PC softmax following Hong et al. (2021) by training the model by ERM for 200 epochs and applying adjustment at testing.

6. **CRT.** We implement CRT following Kang et al. (2019). We train the whole model for 200 epochs and re-train the classifier for 10 epochs with re-weighting loss. We restart the learning rate when re-training the classifier.

7. **LWS.** We implement LWS following Kang et al. (2019). We assign weights to classifiers with learnable scale factors i.e. $\hat{W}_i = f_i \cdot W_i, i \in [L]$. We train the whole model for 200 epochs and then train factors $f_i$ for 10 epochs by re-weighting the loss. We tune the

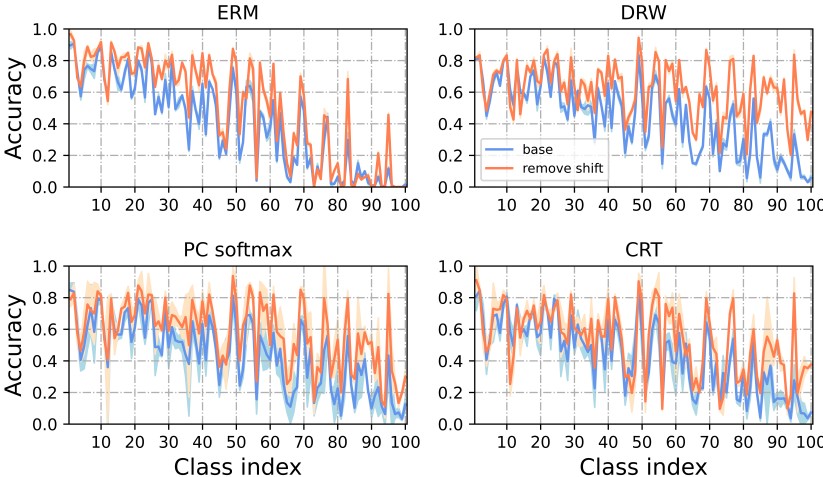

Figure 5: Accuracy on CIFAR100-LT without or with removing CCD shift. It shows similar results that models trained with removing shift sampling get significant performance improvement.

Table 3: Accuracy of different methods on CIFAR-LT without or with removing CCD shift

|  | CIFAR100-LT | | CIFAR10-LT | |
|---|---|---|---|---|
|  | removing shift | base | removing shift | base |
| ERM | 50.7 | 40.28 | 80.37 | 73.24 |
| DRW | 63.97 | 44.41 | 89.01 | 78.04 |
| PC softmax | 58.61 | 44.34 | 87.90 | 79.58 |
| CRT | 52.82 | 43.48 | 87.07 | 77.72 |
| Balance | 66.73 | | 91.53 | |

learning rate on different datasets when learning scale factors as we found it is somewhat sensitive.

## B.2 MORE RESULTS AND ANALYSIS ON CIFAR100-LT

On a whole, results on CIFAR100 agree with those on CIFAR10-LT. In Figure 5 and Table 3, removing shift sampling improves the performance of re-balancing methods and ERM on CIFAR100-LT, which indicates again that shift of CCDs is the key to limit long-tailed recognition.

As for decoupling methods, Table 4 shows re-balanced features outperforms those uniformly trained significantly on two decoupling methods CRT and LWS. It agrees with our explanation of why decoupling methods work: first-stage learning without re-balancing avoids more severe CCD shift while re-balancing with removing shift could benefit feature learning.

Results of logit adjustment on CIFAR100-LT in shown Figure 6. With removing shift sampling $\tau = 1$ is optimal, while without it the best $\tau$ is much bigger than 1. This result again agrees with our supposition that logit adjustment gets sub-optimal in experiments and leaves space to be improved due to CCD shift.

## B.3 CONFIDENCE CALIBRATION AND FEATURE DEVIATION UNDER SHIFT OF CCDS

As two recently studied issues in long-tailed recognition, confidence calibration (Guo et al., 2017) and feature deviation (Ye et al., 2020) have drawn an amount of attention. We conduct experiments to investigate how the shift of CCDs affects these two issues.

Table 4: Accuracy of decoupling methods, CRT and LWS, on CIFAR-LT with different features, without or with removing shift. LWS shows similar results to CRT: re-balance harms feature learning while benefits feature learning with removing shift.

| feature re-balance | classifier adjust | CIFAR100-LT | | CIFAR10-LT | |
|---|---|---|---|---|---|
| | | removing shift | base | removing shift | base |
| - | CRT | 52.82 | 43.48 | 87.07 | 77.72 |
| DRW | CRT | 58.39+5.57 | 42.36-1.12 | 88.42+1.35 | 75.97-1.75 |
| - | LWS | 57.67 | 44.05 | 87.55 | 76.21 |
| DRW | LWS | 61.95+4.28 | 43.63-0.42 | 88.95+1.40 | 75.36-0.84 |
| DRW | | 63.97 | 44.41 | 89.01 | 78.04 |
| RW | CRT | 48.11-4.71 | 28.48-15.0 | 79.51-7.56 | 71.43-6.29 |
| RW | LWS | 49.03-8.64 | 29.76-14.29 | 78.75-8.80 | 71.78-4.43 |
| RW | - | 49.92 | 30.48 | 80.34 | 72.78 |
| Balance | - | 66.73 | | 91.53 | |

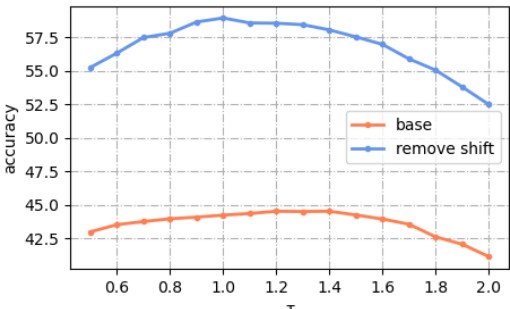

Figure 6: Accury on CIFAR100-LT with varying $\tau$ in post-hoc adjustment. After removing shift $\tau = 1$ gets optimal performance similar to that on CIFAR10-LT.

Table 5: ECE on CIFAR-LT of different methods without or with removing shift sampling.

| | CIFAR10-LT | | CIFAR100-LT | |
|---|---|---|---|---|
| | removing shift | base | removing shift | base |
| ERM | 6.4 | 15.5 | 15.91 | 29.75 |
| DRW | 2.00 | 10.77 | 2.19 | 22.16 |
| PC softmax | 2.55 | 9.25 | 2.73 | 19.43 |
| CRT | 4.40 | 14.54 | 19.5 | 26.63 |
| Balance | 1.77 | | 3.89 | |

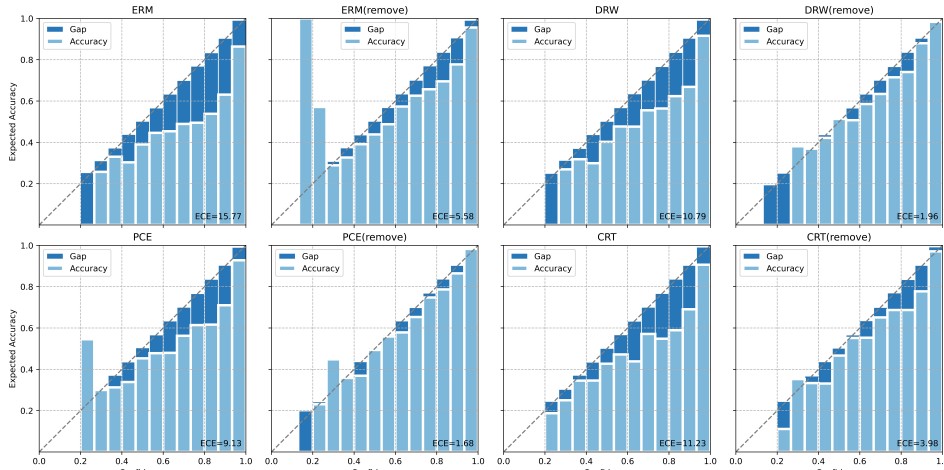

Figure 7: Reliability diagrams of different models trained on CIFAR10-LT. "(remove)" means using removing shift sampling and PCE means PC softmax. The gap between model confidence and error probability is reduced by removing shift sampling significantly.

**Confidence calibration.** Confidence calibration is to make model prediction estimate true correctness likelihood well, which is important in many applications. It has been proved that the calibration of neural networks is generally bad and long-tailed distributions make neural networks even more miscalibrated and over-confident (Guo et al., 2017; Zhong et al., 2021). *Expected Calibration Error* (ECE) is widely used to measure the calibration of a model. With all $N$ instances devided into $B$ interval bins of equal size by their predictions, ECE is calculated as:

$$ECE = \sum_{i=1}^{B} \frac{|S_i|}{N} |acc(S_i) - conf(S_i)|,$$

in which $S_i$ is the set of instances whose predictions fall into the $i$-th bin. $acc(\cdot)$ and $conf(\cdot)$ compute the the accuracy and estimated confidence on $S_i$ respectively. As shown in Table 5, re-balancing methods improve ECEs except CRT on CIFAR-100 and removing shift sampling improves the calibration of different models significantly. However, using both re-balancing methods and removing shift sampling is still worse (in ECE) than that on balanced dataset. We suppose the reason lies in that we still cannot perfectly model a real $P(x|y)$ even with our sampling method. Figure 7 shows the reliability diagrams of different models trained on CIFAR10-LT. With removing shift sampling, the reliability of models is significantly improved.

Table 6: Average feature deviation distance on CIFAR-LT of different methods without or with removing shift sample

|         | CIFAR10-LT | | CIFAR100-LT | |
|---------|----------------|------|----------------|------|
|         | removing shift | base | removing shift | base |
| ERM     | 0.70           | 1.19 | 1.96           | 2.90 |
| DRW     | 0.693          | 1.23 | 2.11           | 3.18 |
| Balance | 0.829          |      | 2.10           |      |

**Feature deviation.** Feature deviation is a phenomenon found in long-tailed recognition by recent works (Ye et al., 2020; 2021). That is the average distance of features learned from long-tailed training dataset usually exhibits imbalance between training and test and the distances of tail classes are larger than those of head classes evidently. Ye et al. (2020) proposes a *feature deviation distance* $dis(j)$ to measure the deviation for class $j$:

$$dis(j; g_\theta) = \frac{1}{R} \sum_{i=1}^{R} \left\| mean(S_K(\{g_\theta(x_{train}^{(j)})\})) - mean(\{g_\theta(x_{test}^{(j)})\}) \right\|_2,$$

in which $\|\cdot\|_2$ is L2-norm, $S_K$ means sub-sampling $K$ instances from class $j$, $R$ is the number of sampling rounds, and $g_\theta$ is the feature extractor. And for convenient comparison, we also use a class-wise mean of feature deviation distance, called *average feature deviation distance*:

$$\overline{dis} = \frac{1}{L} \sum_{j \in [L]} dis(j; g_\theta). \tag{16}$$

Table 6 shows feature deviation is affected by shift of CCDs as removing shift sampling improves this metric on the whole. Surprisingly, the re-balancing method (i.e. DRW) generally intensifies the feature deviation no matter with removing shift or not. Besides, training without re-balancing (i.e. ERM) but with removing shift seemingly even outperforms the balanced result, e.g. 0.70 vs 0.829 on CIFAR10-LT and 1.96 vs 2.10 on CIFAR100-LT. One may conclude that imbalance does not make an obvious difference on feature deviation. In fact, this is because features of head classes benefit from imbalance and removing shift makes up deviated features of tail classes to some extent. As a result, using average feature deviation distance as the measure, imbalance seems not critical for feature deviation.

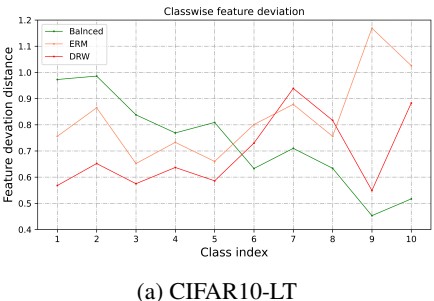
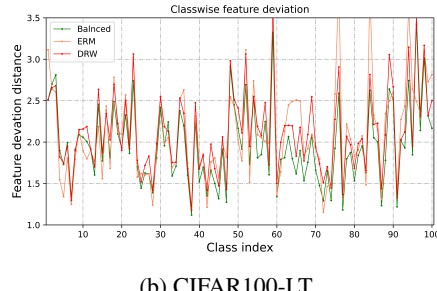

| (a) CIFAR10-LT | (b) CIFAR100-LT |

Figure 8: Class-wise feature deviation on CIFAR10-LT/CIFAR100-LT with removing shift sampling. Even with removing shift sampling and re-balancing it still exhibits imbalanced feature deviation on both CIFAR10-LT and CIFAR100-LT.

Figure 8 shows class-wise feature deviation is affected by imbalance as well. The feature deviation distances of different classes still exhibit imbalance with removing shift. A significant gap of feature deviation distance of tail classes appears between balanced training and re-balanced training with removing shift. We suppose that is because re-balancing methods still leave space to improve feature deviation, e.g. it has been discovered that the imbalanced logit distribution harms feature learning in the beginning stage of training (Ye et al., 2021).

## C   FULL EXPERIMENT DETAILS AND RESULTS

### C.1   EXPERIMENTAL DETAILS

**CIFAR100-LT and CIFAR10-LT.** We apply SGD with batch size 256 and the base learning rate 0.2 to train a ResNet-32 (He et al., 2016) model for 200 epochs, following Hong et al. (2021). We employ the linear warm-up learning rate schedule for the first five epochs and reduce it at epochs 160 and 180 by a factor of 0.01 and use the same weight decay 0.0002 as previous works (Cao et al., 2019; Zhou et al., 2022).

**Tiny-ImageNet-LT.** Following Park et al. (2021), we apply SGD with batch size 128 and weight decay 0.0002 to train a ResNet-18 model for 100 epochs. We set the base learning rate to 0.1 and reduce it at epochs 50 and 90 by a factor of 0.1.

**ImageNet-LT.** Following Hong et al. (2021), we apply SGD with batch size 128 and weight decay 0.0005 to train a ResNet-50 model for 90/180 epochs. We perform a cosine learning rate scheme with an initial learning rate of 0.05.

**Reproducing baseline results.** We combine our DRA with various existing methods and compare their results with these baselines methods in long-tailed recognition. For the baselines combined with DRA, we implement them based on their official codes with default parameters in their modules

under our training setting for fair comparison, such as model architecture and optimization hyperparameters. There are a few results disagreeing with original numbers reported in their papers, caused by their different training settings from ours, e.g. we get 44.34 top-1 accuracy of PC softmax on CIFAR100-LT while the reported number is 45.3 in Hong et al. (2021), which is because they use bigger weight decay parameters and only perform experiments under one random seed. For other baselines, we reproduce the results using publicly available official codes or direct use numbers reported in the original papers. For all results we implement or reproduce based on official code, we run five times with different seeds and report the average.

## C.2 IMPLEMENTION OF DRA

---

**Algorithm 1** DRA: augmenting with a sequence of examples

---

**Input:** $\{(x_i, y_i)_{i=1}^{N_{batch}}\}$, a batch of instances
**Input:** $\{\lambda_j\}_{j \in [L]}$, the class-aware multipliers
**Output:** $Augbatch = \{(x_i^{aug}, y_i)\}_{i=1}^{N_{batch} \cdot (M-k)}$
 1: $Augbatch \leftarrow \{\}$
 2: **for** $i = 1, \ldots, N_{batch}$ **do**
 3:     $x_i^0 \leftarrow x_i, j \leftarrow 0, g_i^* \leftarrow (2\epsilon, 0, \cdots, 0)$
 4:     **while** $j < M$ and $\|g_i^*\| > \epsilon$ **do**
 5:         $g_i^* = \nabla_x \lambda_{y_i} c((x_i^j, y_i), (x_i, y_i)) - l_\theta(x_i^j, y_i);$   ▷ Compute gradient of inner-optimization
 6:         $x_i^{j+1} \leftarrow x_i^j - \alpha_{inner} \cdot g_i^*$          ▷ Update augmentation examples by gradient
 7:         **if** $j \geq k$ **then**
 8:             $Augbatch = concate((Augbatch; (x_i^j, y_i)))$        ▷ Save a sequence of exampls
 9:         $j = j + 1$
10: **return** $Augbatch$          ▷ Return augmented instances for outer-optimization

---

---

**Algorithm 2** WRM (Sinha et al., 2017): augmenting with the last example

---

**Input:** $\{(x_i, y_i)_{i=1}^{N_{batch}}\}$, a batch of instances
**Input:** $\{\lambda\}$, the multiplier
**Output:** $Augbatch = \{(x_i^{aug}, y_i)\}_{i=1}^{N_{batch} \cdot (M-k)}$
 1: $Augbatch \leftarrow \{\}$
 2: **for** $i = 1, \ldots, N_{batch}$ **do**
 3:     $x_i^0 \leftarrow x_i, j \leftarrow 0, g_i^* \leftarrow (2\epsilon, 0, \cdots, 0)$
 4:     **while** $j < M$ and $\|g_i^*\| > \epsilon$ **do**
 5:         $g_i^* = \nabla_x \lambda c((x_i^j, y_i), (x_i, y_i)) - l_\theta(x_i^j, y_i);$     ▷ Compute gradient of inner-optimization
 6:         $x_i^{j+1} \leftarrow x_i^j - \alpha_{inner} \cdot g_i^*$          ▷ Update augmentation examples by gradient
 7:         $j = j + 1$
 8:     $Augbatch = concate((Augbatch; (x_i^j, y_i)))$          ▷ Save the last exampl
 9: **return** $Augbatch$          ▷ Return augmented instances for outer-optimization

---

The data augmentation process of DRA is summarized in Algorithm 1. Noting that the class-ware multiplier $\lambda_j$ has a negative correlation with the radius of uncertainty set so a positive correlation with the number of instances of class $j$, thus we set $\lambda_j = normalize\{N_j{}^\beta\} * S$ with $\beta, S \geq 0$ and $normalize$ meaning scale the vector to unit vector. $\beta$ determines robustness difference over classes and $S$ determines overall robustness level. In this way, we could reduce the multipliers of DRA to only two hyperparameters. We set $M = 10$ and $\alpha_{inner} = 0.1$ in DRA. As for the hyperparameter $k$, the number of iterations starting to be used to augment examples, we set it as half of the whole number of iterations $M$ heuristically or $M - 1$, which is equivalent to directly utilizing WRM as the solution of our primal problem. We do not fine-tune $k$ to get an optimal number but only select from the above two choices, which, however, gets pretty good result.

Inspired by (Kim et al., 2020; Samuel & Chechik, 2021), we apply DRA in the later stage of training since we need an initial model for DRA to generate augmentation examples. Specifically, for

CIFAR10-LT and CIFAR100-LT we start to use DRA on 160 epochs, for Tiny-ImageNet-LT we start to use DRA on 90 epochs. For ImageNet-LT we start to use DRA on 80 epochs when training for 90 epochs while we start to use DRA on 150 epochs when training for 180 epochs. And for CRT, we only use DRA in the training of the second stage.

Following Hong et al. (2021); Kim et al. (2020), we tune the hyperparameters of DRA by grid search on the validation set. WRM uses the following strategy to determine the hyperparameter multiplier: $\lambda = C \cdot E_{P_N(x)}[\|x\|_2]$ while $C = 0.04$ is a constant and $E_{P_N(x)}[\|x\|_2]$ is the average norm of all instances in training set. However, this strategy is not helpful in determining the hyperparameters $S$. For example, $E_{P_N(x)}[\|x\|_2] = 248.39$ on CIFAR10-LT while it is $222.42$ on CIFAR100-LT. The two numbers are similar but the optimal $S$ found by our grid search on the two sets disagrees a lot. Moreover, scaling the optimal $S$ on CIFAR10-LT by the proportion of $E_{P_N(x)}[\|x\|_2]$ does not work well either. How to determine the hyperparameters for DRA more elegantly could be a future research point to further improve our method.

## C.3 RESULTS ON IMAGENET-LT

Table 7: The performances on ImageNet-LT. We report accuracy on three splits of classes: Many-shot (more than 100), Medium-shot (20-100) and Few-shot (less than 20), following Hong et al. (2021); Zhong et al. (2021). The small red font denotes performance gain. ‡ means results from the original paper.

| Method | Many | Medium | Few | All |
|---|---|---|---|---|
| *90epochs* | | | | |
| ERM | **65.1** | 35.7 | 6.6 | 43.1 |
| Decouple-CRT | 61.22 | 47.52 | 26.41 | 49.52 |
| Decouple-$\tau$-norm‡ | 59.1 | 46.9 | **30.7** | 49.4 |
| Decouple-LWS‡ | 60.2 | 47.2 | 30.3 | 49.9 |
| PC softmax | 60.4 | 46.7 | 23.8 | 48.9 |
| Logits Adjustment | 60.62 | 47.33 | 27.53 | 49.25 |
| LADE | 60.34 | 47.37 | 27.82 | 49.20 |
| Decouple-CRT-DRA | 61.63 | 47.53 | 30.61 | **50.25**+0.72 |
| PC softmax-DRA | 60.38 | 46.81 | 24.23 | 49.11+0.21 |
| Logits Adjustment-DRA | 60.63 | **47.84** | 27.60 | 49.49+0.24 |
| *180epochs* | | | | |
| ERM | **66.84** | 40.89 | 11.54 | 46.05 |
| Decouple-CRT | 61.59 | 47.70 | 30.32 | 50.3 |
| PC softmax | 62.13 | 49.25 | 30.51 | 51.18 |
| Logits Adjustment | 62.67 | **49.96** | 31.67 | 51.90 |
| LADE | 62.80 | 49.76 | 33.4 | 52.14 |
| Decouple-CRT-DRA | 61.81 | 49.43 | 31.57 | 51.37+1.07 |
| PC softmax-DRA | 62.41 | 49.72 | 31.89 | 51.64+0.46 |
| Logits Adjustment-DRA | 63.29 | 48.88 | **32.00** | **52.32**+0.42 |

## C.4 MORE ABLATION STUDY

### C.4.1 COMPARISON WITH DATA AUGMENTATION METHODS

We can regard our DRA as generating examples for data augmentation during model training and so does WRM. Hence, it is natural to ask what is the relationship between DRA and other data augmentation methods in long-tailed recognition. In our view, DRA focuses on assigning robustness over classes against unreliable CCD estimated from scare instances, while other data augmentation methods aim to improve the diversity of instances heuristically, e.g. through head-to-tail transfer or instance generation. Moreover, those data augmentation methods neither attend to the key issue of CCD shift nor provide theoretical guarantee. In our experiments, we conduct a comparison with two recent works GIT and M2m (Zhou et al., 2022; Kim et al., 2020) with respect to performance

and training cost. Besides, for augmentation methods that are adopted in balanced learning, we suppose they are complementary to DRA. So we also conduct an experiment of combining DRA with a representative augmentation method Mixup (Zhang et al., 2017).

**Performance comparison with learned augmentation methods.** To make a fair comparison with GIT and M2m, we conduct experiments under the same settings of GIT and M2m. As the official code of GIT and M2m cannot work well with our larger batchsize setting, we keep all settings except changing the batchsize to 128 and initial learning rate to 0.1.As shown in Table 8, generally, DRA gets comparable performance to M2m and GIT with a small gap (less than 0.2%) with CE-DRS on CIFAR10-LT and outperforms them on all other settings. It is surprising WRM does not get any significant improvement on all baselines, which shows assigning the same robustness to all classes is not appropriate as much robustness is unnecessary for head classes and may cause over-pessimism while little robustness cannot improve generalization for tail classes evidently. In contrast, DRA is more theoretically sound and more stable than these heuristic augmentation strategies. We hope this could encourage future methods to solve the shift of $P(x|y)$ in long-tailed recognition more effectively.

Table 8: Comparision of other augmentation methods with DRA. ‡ means reproduced result from official code. Best results are marked in bold.

| Method | Aug. strategy | CIFAR10-LT | CIFAR100-LT |
|---|---|---|---|
| CE-DRS | - | 75.87 | 41.21 |
| | GIT‡ | 77.06 | 41.86 |
| | M2m‡ | **77.5** | 42.2 |
| | WRM | 75.92 | 41.31 |
| | DRA | 77.38 | **42.61** |
| LDAM-DRS | - | 77.47 | 42.78 |
| | GIT‡ | 78.49 | 43.49 |
| | M2m‡ | 78.68 | 43.12 |
| | WRM | 77.34 | 43.11 |
| | DRA | **78.76** | **43.53** |

**Training cost of DRA compared with learned augmentation methods.** We compare DRA with learned data augmentations e.g. GIT and M2m with respect to the training cost in Table 9. All the methods are measured on an NVidia card (GTX 2080Ti). The inner optimization of DRA increases the training cost exactly as it needs to make the model forward and backward to compute gradients on the input to generate examples. Considering all these methods are only used in the later phase of training e.g. the last 40 epochs of all 200 epochs on CIFAR-LT, they are not quite time-consuming in the end-to-end training part. However, GIT and M2m need much additional time to train a model as preparation e.g. a GAN for GIT and a network classifier trained by ERM for M2m. DRA avoids the additional time and just slightly increases the training time per epoch, where the gap between DRA and GIT or M2m could be ignored considering their additional time cost for preparation. Besides, it seems using a sequence of examples i.e. $k = M/2$ does not cost more training time than using the last example.

**Combination with Mixup.** We also explore whether DRA is compatible with existing general data augmentation methods or not. We use the simple, efficient and widely-used Mixup(Zhang et al., 2017) as an example. Specifically, we only use Mixup to pre-train the model before using DRA. Specifically, on CIFAR-LT, we use Mixup in the early 160 epochs and DRA in the later 40 epochs of 200 epochs in training. We also compare ours with state-of-the-art Mislas(Zhong et al., 2021) based on Mixup. Table 10 shows DRA is suitable to train with Mixup as DRA even further improves the performance which has already been enhanced by Mixup and outperforms Mislas, showing compatibility between the two methods. It implies that even if data augmentation methods would improve the diversity of data, they are still possibly biased due to unreliable empirical distribution and thus could enjoy a "correction" from DRA.

Table 9: Comparison of training cost of CE-DRS with different augmentations on CIFAR100. The training time is counted in seconds. GIT and M2m need additional time to train another model, e.g. a GAN and a model trained by ERM respectively. Whereas, DRA just slightly increases training time.

|  | Aug. strategy | Time per epoch(s) | Additional time(s) |
|---|---|---|---|
| CE-DRS | - | 3.72 | - |
|  | GIT | 25.84 | 33710.41 |
|  | M2m | 29.25 | 669.2 |
|  | DRA($k = M - 1$) | 27.60 | - |
|  | DRA($k = M/2$) | 32.26 | - |

Table 10: Ablation results on combining Mixup and DRA, ‡ means reproduced result from official code. Best results are marked in bold. The small red font denotes performance gain from our DRA.

| Method | Aug. strategy | CIFAR10-LT | CIFAR100-LT |
|---|---|---|---|
| CE-DRS | Mixup | 79.70 | 47.08 |
|  | Mixup+DRA | 80.59₊₀.₈₉ | **47.39**₊₀.₃₁ |
| PC softmax | Mixup | 81.29 | 46.12 |
|  | Mixup+DRA | **82.41**₊₁.₁₂ | 46.35₊₀.₂₃ |
| Mislas‡ | - | 82.10 | 47.19 |

### C.4.2 DRA ON CONFIDENCE CALIBRATION

Inspired by the discovery in our empirical study that confidence calibration is effected by both imbalance and shift of CCDs, we expect DRA would relieve over-confidence of models. From our experiments, it is kind of surprising that DRA not only improves calibration but also improves the well-calibrated models by Mixup most of the time. We use ECE (Guo et al., 2017) as the measure.

In Figure 9, it seems just with re-balancing methods model cannot be calibrated well and DRA actually gets smaller ECE on these methods, which validates again that confidence calibration is affected by shift of CCDs and DRA improves it by making the model more robust to the shift.

Besides, considering the discovery in recent works (Zhong et al., 2021; Xu et al., 2021) that Mixup has a significant positive effect on calibration, we conduct experiments to measure ECE with both Mixup and DRA. Figure 10 shows DRA further improves calibration on the basis of Mixup most of time. The only exception is CE-DRS on CIFAR10-LT, in which DRA weakens calibration but boosts its accuracy as shown in Table 10. It seems with both Mixup and DRA the regularization on logits is too strong to obtain good calibration as the model gives confidence even lower than the accuracy instead of usual over-confidence on neural network (Guo et al., 2017). These results also raise an interesting question: Do calibration and accuracy agree in long-tailed recognition? Or more specifically, does good calibration mean a good model in long-tailed recognition (Xu et al., 2021)? Actually, our experiment shows that it is possible to boost performance while weakening calibration.

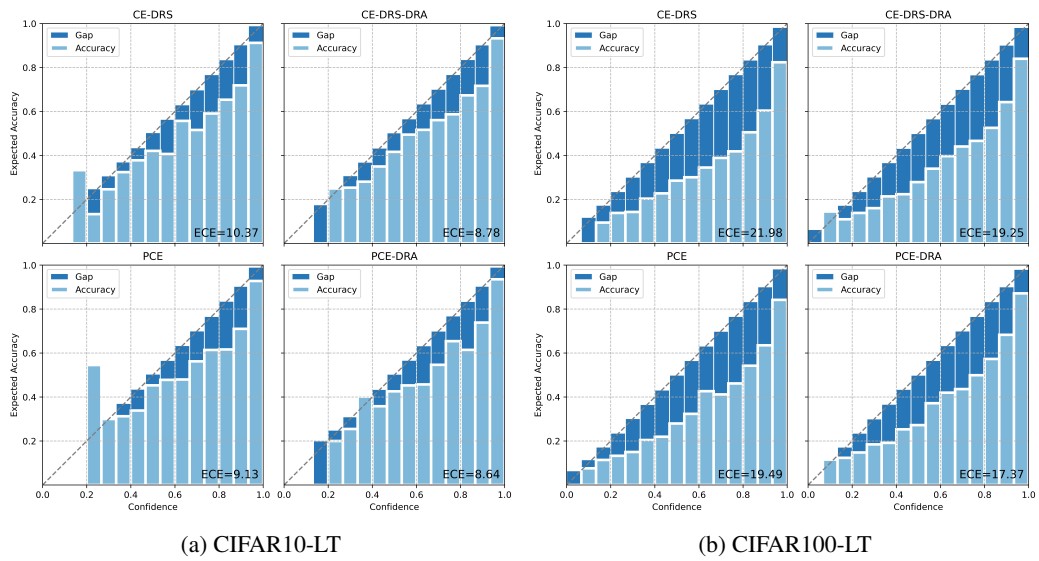

(a) CIFAR10-LT                    (b) CIFAR100-LT

Figure 9: ECE (%) and reliability diagram on CIFAR10-LT/CIFAR100-LT. DRA gets better calibration performance marginally. PCE means PC softmax.

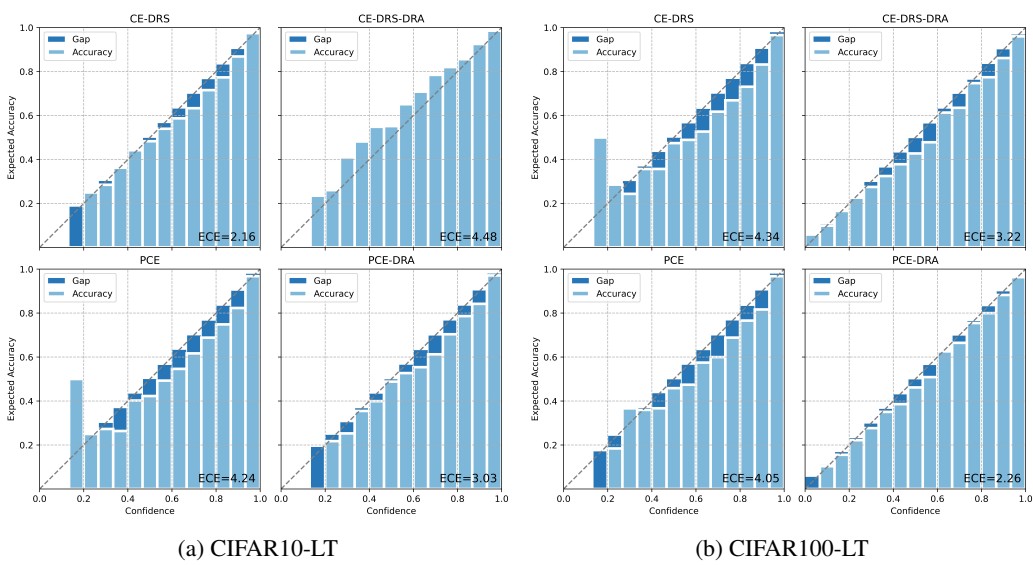

(a) CIFAR10-LT                    (b) CIFAR100-LT

Figure 10: ECE (%) and reliability diagram on CIFAR10-LT/CIFAR100-LT with Mixup. DRA even improves the calibration of models with Mixup, which have obtained small ECE and been calibrated well. PCE means PC softmax.

### C.4.3 VISUALIZATION OF AND DISCUSSION ON EXAMPLES GENERATED BY DRA

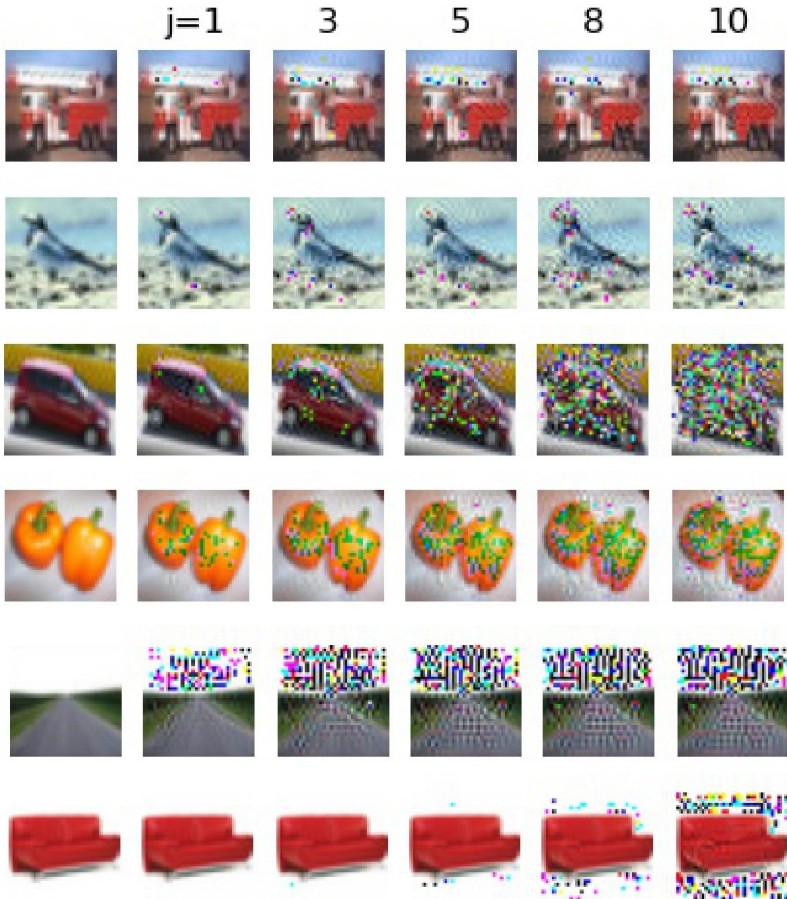

Figure 11: Examples generated by DRA on CIFAR-LT. Left most column are original examples and $j$ is the number of iterations. DRA tends to add pixel-level transformations to obtain harder examples while some of these transformations agree with semantic information.

To get more precise understanding, we visualize the examples generated by DRA, as in Figure 11. It exhibits that DRA adds pixel-wise transformation to instances in order to generate harder examples as data augmentation. Some of these transformations tend to erase conspicuous features in an instance by adding "noise". In this way, model is encouraged to extract broader features instead of overfitting features it has learned. For example, DRA seems to add "noise" to erase the ladder from "fire engine" in the first row. As ladder is a salient feature for fire engine, and this augmentation makes model attend to more recognizable features e.g. water tank and red color. Similarly, in the second row, DRA adds "noise" on the beaks and claws of the bird and encourages the model to take in more features to help recognition instead of over-depending on the two features.

In addition, more surprisingly, DRA can catch semantic information while discard nuisance information by itself. As in the fourth row, the yellow sweet peppers happen to be transformed to green while pixels out of the peppers almost kept unchanged. In other words, DRA makes a transformation of color, and the transformed instance still belongs to "sweet pepper" class. In this way, DRA performs transformations that keep the semantics unchanged. As shown in the bottom two rows, DRA could give transformations on the background (nuisance information) and keep semantic information: almost all of the pixel-wise transformations in the last but one row lie in the sky, and the semantic object "road" isn't changed. In the last row, the pixels of sofa are not transformed while the background is changed.

From these results, we infer that seeking harder examples for a robust model sometimes agrees with transformations that keep the semantics of instances. For example, background intensity could serve as examples to obtain distributional robustness under a few situations, and this can be an explanation of why GIT improves performance by adding transformations keeping semantics of instances. However, when transformations that keep semantics are not that effective to gain distributionally robustness, DRA tends to seek other examples with obvious features are erased.

From the observation that examples from DRA do not have to keep semantics and seem like instances with "noise", we hope DRA could suggest a new motivation for data augmentation: making a more robust model for unknown distribution shift instead of utilizing information to reduce the shift, e.g. GIT, ISDA (Zhou et al., 2022; Wang et al., 2019; Li et al., 2021). This motivation, making a model robust to shift, could give hints to explain the rationality of existing data augmentations that don't have to keep semantics, e.g. adding pure noise to instances as data augmentation (Zada et al., 2022) and those based on Mixup (Zhang et al., 2017; Chou et al., 2020; Zhong et al., 2021; Xu et al., 2021).

