# OpenReview forum: "Beyond re-balancing: distributionally robust augmentation against class-conditional distribution shift in long-tailed recognition"
_ICLR.cc/2023/Conference — Submitted to ICLR 2023_

### Official Review · Reviewer_9ohR · 2022-10-22

**Confidence:** 4
**Clarity, Quality, Novelty And Reproducibility:** No code is provided.
**Correctness:** 3
**Technical Novelty And Significance:** 3
**Empirical Novelty And Significance:** Not applicable
**Recommendation:** 5

**Strength And Weaknesses:**

Strength:

(1) The paper gives a sound theoretical analysis.


Weakness:

(1) Experiments are conducted only on CIFAR-LT or TinyImageNet-LT, which makes the experimental results unconvincing. In the long-tailed community, we usually validate the effectiveness of methods with ImageNet-LT, iNaturalist 2018, and Places-LT.

(2) The authors claim that "decoupling methods avoid more severe CCD shift". Does it mean that it is necessary to address long-tailed recognition with 2 stages? However, recent state-of-the-art methods, like PaCo [1], train representation and classifier jointly. Can the proposed method improve performance when it is applied to such a strong baseline?

[1] Parametric Contrastive Learning. ICCV 2021.

(3) How can we remove the effect of shifted CCD?
     As analyzed in Sec 3.2, function (5) is used to sample images. From my understanding, with function (5), all images can be viewed by the trained model, which is the same as that of training models on balanced data. The only difference is that the sample probability for tail-class images will be smaller.

(4) Analysis of the computational cost for the proposed method is missing.





**Summary Of The Paper:**

(1) This paper aims to solve the long-tailed recognition task.
(I) This paper proposes that the class-conditional distribution shift can not be ignored in practice with empirical analysis.
(II) Based on this observation, this paper proposes to utilize DRO to minimize the worst risk and It can be understood as an adaptive data augmentation method.
(2) Experiments on CIFAR-LT and Tiny-ImageNet-LT are conducted. Some improvements are observed compared with baselines.


**Summary Of The Review:**

The paper aims to address the long-tailed recognition problem. Analysis of the CCD issue is interesting.
However, the evaluation is unconvincing. Comparisons with state-of-the-art methods are missed. Also, only small datasets are evaluated.

If concerns are well addressed, I'm glad to raise my score.

---

> ### Author Response · Authors · 2022-11-15
> **Response to reviewer 9ohR(1/2)**
>
> Thank you for your insightful comments and suggestions.
> > Experiments are conducted only on CIFAR-LT or TinyImageNet-LT, which makes the experimental results unconvincing.
>
> Besides the experiments on CIFAR-LT and TinyImageNet-LT, we also conduct experiments on the large-scale dataset ImageNet-LT (results in Table 7, Appendix C.3). The results show that DRA improves re-balancing baselines e.g. CRT, PC softmax and logits adjustment on ImageNet-LT as well, and beats prior SOTA LADE, which verifies the effectiveness of proposed DRA. We hope this could address your concerns.
> > Does it mean that it is necessary to address long-tailed recognition with 2 stages? However, recent state-of-the-art methods, like PaCo [1], train representation and classifier jointly.
>
> Our analysis of decoupling methods [2] provides an explanation on the counter-intuitive fact that re-balancing generates worse feature, that is,  re-sampling and reweighting exacerbate CCD shift of tail classes.  Then a new issue emerges: how to avoid exaggerating unreliable CCD of tail classes when applying re-balancing methods. We give new insight to the effectiveness of decoupling methods: it avoids the two-edged sword effect of re-balancing by only using class-balanced sampling in the second stage. Hence, **the key point of decoupling is not enforcing a two-stage training but avoiding worse features from class-balanced sampling which exacerbates CCD shift**. Decoupling is a solution for this purpose but not the only one.
>
> Besides, we are happy to agree that self-supervised learning and contrastive learning may be used to generate better features as well. As many previous works [3,4,5,6] present, self-supervised learning and contrastive learning benefit long-tailed learning. Recent works find contrastive learning could obtain more balanced feature space [3] and learn richer features from frequent data that are transferable to rare data [4].
>
> For PaCo [1], as above analysis, it is reasonable to suppose that **contrastive learning of PaCo benefits the feature learning under its re-balanced strategy following logit adjustment (Balanced softmax)[7.8]**, so it achieves good performance without two-stage training.
>
> [1] Parametric Contrastive Learning. ICCV 2021.
> [2] Decoupling representation and classifier for long-tailed recognition. ICLR 2020.
> [3] Exploring Balanced Feature Spaces for Representation Learning. ICLR 2021.
> [4] Self-supervised Learning is More Robust to Dataset Imbalance. ICLR 2022.
> [5] Rethinking the Value of Labels for Improving Class-Imbalanced Learning. NeurIPS 2020.
> [6] Targeted Supervised Contrastive Learning for Long-Tailed Recognition. CVPR 2022.
> [7] Long-tail learning via logit adjustment. ICLR 2021.
> [8] Balanced Meta-Softmax for Long-Tailed Visual Recognition. NeurIPS 2020.

---

> ### Author Response · Authors · 2022-11-15
> **Response to reviewer 9ohR(2/2)**
>
>
> > Can the proposed method improve performance when it is applied to such a strong baseline?
>
> As above discussion, considering that contrastive learning benefits feature learning and the worse feature from re-balancing is due to CCD shift, it is natural to ask if contrastive learning could address CCD shift. However, identifying the effect that contrastive learning makes on CCD shift and applying DRA to contrastive learning are both non-trivial. Even if DRA is applied rigidly, how to apply various data augmentation methods for encoder and momentum encoder in contrastive learning is still a key issue. More critically, even data augmentation, e.g. RandAugment [9], having been proved effective on some other methods, may not be suitable for contrastive learning with some strategies[1,10]. Therefore, whether it can be applied to contrastive learning may not be a good metric for data augmentation methods. Considering our analysis and proposed DRA are based on supervised cross-entropy learning, the above issues are out of scope of this work. On the whole, we agree that addressing CCD shift by contrastive learning is reasonable and interesting for future work.
>
> Besides, to show the ability of DRA to improve stronger baselines, we choose a data augmentation method Mixup [11] to be allied with DRA, which is widely-used and effective on long-tailed learning and there are several state-of-the-art methods [12,13] based on it.
> | Method   |      Aug.strategy      |  CIFAR10-LT | CIFAR100-LT |
> |:----------|:-------------|:------|:------|
> | CE-DRS|  mixup | 79.7|47.08|
> | |    mixup+DRA   |   80.59 |**47.39**|
> | PC softmax|  mixup | 81.29 |46.12|
> | |    mixup+DRA   |   **82.41** |46.35|
> |Mislas |   -   |   82.10 |47.19|
>
> The above table is also posted as Table 10 in Appendix C.4.
>
> It shows that DRA further improves the performance of Mixup and outperforms Mislas [12], which is a state-of-the-art method using mixup as well, which proves DRA has the potential to improve strong baselines. It also implies that even if these methods have superior performance with improved data diversity, they are still possibly biased due to unreliable empirical distribution
> and thus could enjoy a “correction” from DRA. Please refer to Appendix C.4 and Table 10 for more clarification.
>
> We also conducted experiments for confidence calibration of DRA in Appendix B.3, in which DRA could obtain even better confidence calibration on models trained with mixup, which has been calibrated well.
>
> [9] Randaugment: Practical automated data augmentation with a reduced search space. NeurIPS 2020.
> [10] Contrastive learning with stronger augmentations. IEEE TPAMI 2022.
> [11] mixup: Beyond empirical risk minimization. ICLR 2018.
> [12] Improving Calibration for Long-Tailed Recognition. CVPR 2021.
> [13] Towards Calibrated Model for Long-Tailed Visual Recognition from Prior Perspective. NeurIPS 2021.
> > How can we remove the effect of shifted CCD?
>
> It is worth emphasizing that our proposed DRA does not aim at "removing shift" as it is intractable and almost impossible to restore the real CCD from scarce instances when learning from a long-tailed distribution. Our idea is **training a model robust to possible CCD shift**. The proposed DRA validates the effectiveness and rationality of our idea.
>
> Please be noted that the sample distribution in (5) is not a part of our proposed method to address CCD shift. Instead, **it is used as an oracle in the empirical study in Section 3 to show the influence of CCD shift**. With the same imbalanced label distribution $P_{LT}(y_j)$, a more reliable CCD $P_{N_j,bal}(x|y_j)$ (than $P_{N_j,LT}(x|y_j)$) from the balanced distribution is utilized as an oracle estimation of real CCD. Please refer to Appendix B.1 "Assumption of removing shift" for more clarification.
> > Analysis of the computational cost for the proposed method is missing.
>
> We conduct experimental comparisons of training cost on CIFAR-100-LT with similar data augmentation methods that tend to alleviate the scarcity of tail instances e.g. GIT[14],M2m[15]. The results are shown in the following table, which is also posted as Table 9 in Appendix C.4.
> | Method | Aug.strategy |  Time per epoch | Additional time(s)|
> |:------|:---------|:-----|:------|
> | CE-DRS| -        | 3.72 |-      |
> |       |GIT       |25.84 |33710.4|
> |       |M2m       |29.25 |669.2  |
> |       |DRA(k=M-1)|27.60 |-      |
> |       |DRA(k=M/2)|32.26 |-      |
>
> For training time per epoch, DRA is comparable to prior methods. Considering DRA does not need additional training time and is only used in the late phase of training (e.g. last 40 epochs of 200 epochs on CIFAR-LT, last 10 epochs of 90 epochs on ImageNet-LT), it has much less training cost than prior data augmentation methods. Please refer to Appendix C.4 for more clarification.
>
> [14] Do deep networks transfer invariances across classes? ICLR 2022.
> [15] M2m: Imbalanced classification via major-to-minor translation. CVPR 2020.

---

### Official Review · Reviewer_7ExN · 2022-10-23

**Confidence:** 4
**Correctness:** 2
**Technical Novelty And Significance:** 2
**Empirical Novelty And Significance:** 2
**Recommendation:** 3

**Clarity, Quality, Novelty And Reproducibility:**



**Writing suggestions**:
- As mentioned before, the exposition of Section 4 can be significantly improved.
- WRM is discussed in the paper (specifically while introducing DRA in Sec 4.2) without really explaining what the method is. For completeness, it would be great if the authors can add a brief discussion of this method.
- Minor: In the conclusion paragraph, "we" is capitalized without any reason in the first line.
- Statements like this "The convergence and generalization of DRA are theoretically guaranteed." are misleading in the abstract. Since authors do not present an end-to-end theoretical guarantee that DRA would generalize in all settings without any assumptions, it would be great to contextual and tone down the language of the contribution.


**Missing related work**:
- The assumption of $p_{train}(x|y) = p_{test}(x|y)$ and $p_{train} (y) \ne p_{test}(y)$ is well known as label shift assumption in the DA literature [1,2,3,4]. Authors should add a discussion on this line of literature with appropriate references.

[1] Z. C. Lipton, Y.-X. Wang, and A. Smola. Detecting and Correcting for Label Shift with Black Box Predictors. In International Conference on Machine Learning (ICML), 2018.

[2] S. Garg, Y. Wu, S. Balakrishnan, and Z. C. Lipton. A unified view of label shift estimation. In Advances in Neural Information Processing Systems (NeurIPS), 2020.

[3] A. Storkey. When Training and Test Sets Are Different: Characterizing Learning Transfer. Dataset Shift in Machine Learning, 2009

[4] M. Saerens, P. Latinne, and C. Decaestecker. Adjusting the Outputs of a Classifier to New Priori Probabilities: A Simple Procedure. Neural Computation, 2002.



**Strength And Weaknesses:**

**Strengths**:
- The paper tackles an important problem of long-tailed recognition.
- Empirical results with the proposed DRA technique in Table 2 are promising.


**Weaknesses**:
- Discussion on Proposition 1 is a bit misleading. Authors argue that ` with enough instances from each class, models working well on training set could perform well in the test'. However, it is well accepted that usually even for balanced benchmark datasets (i.e., CIFAR100 or Imagenet), with overparameterized networks, we can not approximate $p(x|y)$ or the generalization performance. In general, for high-dimensional vision datasets typically used, the sample size is small enough that we would see high drift as evaluated by Proposition 1. If this understanding is incorrect, I would encourage authors to provide examples in balanced datasets like CIFAR10, CIFAR100, or Imagenet where bounds like in Proposition 1 are tight.
- To put it differently, the issue here is that authors are trying to argue that it is not possible to estimate $p(x|y)$ from finite samples in LT classes. While this is understandable, it is unclear if (i) any method actually explicitly computes $p(x|y)$ to handle long tail classes; and (ii) $p(x|y)$ can be estimated reasonably even for balanced benchmark datasets like CIFAR-10, CIFAR-100, etc.
- In light of the above comment, I guess a more appropriate argument would be to discuss the relative sample sizes in tail classes of long-tailed datasets versus sample sizes in balanced datasets like CIFAR100. However, I am not sure if such an argument could be theoretically made with bounds in Proposition 1.
- It is unclear if equation (4) is actually used to sample datasets in prior work. To the best of my understanding equation (4) represents the empirical distribution of a dataset sampled to satisfy LT phenomena. The core issue here is if we use sampling with replacement from distribution in (4) versus distribution in equation (5), we may see a lot of repeated examples from long tail classes in case of (4) than in case of (5). While this is intuitive, it is unclear if prior work has really used sampling with replacement from a model as in (4). Can authors add references to prominent prior works which used this strategy for benchmarking?
- The paper is hard to follow starting from Section 4. In particular, I could not find a clear description of the proposed DRA method in the main paper. In Algorithm 1, it is unclear why the loop in Line 4 will be executed with $g_i^*$ initialized the way it is in Line 3.
- Figure 3 is also a bit unclear. It is hard to follow the description of the DRA method from this figure. For example, it is unclear what "From the k-th,...,M-th loops" mean? What is the difference between blue and orange arrows? Why there are two input images to the networks (one in blue and one in orange)?

**Summary Of The Paper:**

In this work, the authors tackle the problem of long-tailed recognition. The authors highlight the issue of Class-Conditional Distribution (CCD) shift due to scarce instances. The authors propose an adaptive data augmentation method, Distributionally Robust Augmentation (DRA) to improve performance for long-tailed recognition tasks. Experiments across several semi-synthetic datasets illustrate the efficacy of DRA.

**Summary Of The Review:**

Overall, I think while this paper tackles an important problem of long-tailed recognition, the paper is hard to follow in its current stage and several arguments are presented without concrete theoretical and empirical evidence. In my understanding, the motivation for the CCD shift can be presented in a simpler way. A clear description of the proposed DRA algorithm is also missing in the paper. While the proposed DRA taking looks promising from empirical results, the connection with the proposed motivation is a bit thin.

I encourage authors to participate in the discussion to clarify if I share any misunderstanding. I will be open to changing my score if they think I misunderstood any of the key arguments made in the paper.

---

> ### Author Response · Authors · 2022-11-15
> **Response to reviewer 7ExN(1/2)**
>
> We thank the reviewer for the careful and insightful comments and suggestions.
> >Concers of Proposition 1: the first three reviews in weaknesses.
>
> We appreciate the careful and constructive comments from the reviewer for Proposition 1 again, which is critical to our empirical study of CCD shift.
>
> In our understanding, the main concern of the reviewer for Proposition 1 is that in our empirical study, CCD from balanced distribution $P_{N_j,bal}(x|y_j)$ serves as an oracle estimation of real CCD $P(x|y_j)$, but the gap between the two remains unknown. If applying the bound of Proposition 1, how fast the bound decreases with the increase of the sample size need to be computed.
>
> The bound of Proposition 1 is actually intractable in practice since the real CCD $P(x|y)$ cannot be computed from finite sample. Besides, **it is possible that the oracle $P_{N_j,bal}(x|y_j)$ actually has a considerable gap to the real CCD $P(x|y_j)$**, that is why the generalization on balanced datasets (with much more samples than long-tailed datasets) may still require large sample size.
>
> However, we argue that **Proposition 1 can still give useful insights and support our empirical study**.  Using $P_{N_j,bal}(x|y_j)$ as an oracle is reasonable in our experiments despite it will fail in some rare case. In fact, our empirical study aims at identifying the CCD shift and the influence it makes on previous methods. In other words, **we pursue proving how unreliable and influential the CCD in long-tailed recognition is instead of how good the CCD estimated from balanced training set is**.
>
> Let us explain more from the reverse. Assume the CCD from long-tailed distribution $P_{N_j,LT}$ has been relatively reliable, i.e. the left term of Proposition 1 is small. In that case, using $P_{N_j,bal}(x|y_j)$ would not give significant improvement, and a tighter bound computed with larger $N_j$ will not help, as presented in Proposition 1. Similarly, if CCD shift is not blamed for the counter-intuitive facts of decoupling methods and logit adjustment, using oracle will not either. Therefore, **as long as the oracle $P_{N_j,bal}(x|y_j)$ brings significant performance improvement, we can identify the CCD shift**.
>
> In our empirical study, with sample distribution (5), the results are improved significantly (Figure 1) and the counter-intuitive facts do not appear (Figure 2, Table 1), which suggests that CCD from long-tailed distribution is unreliable and CCD shift influences the performance a lot.
>
> **This way to identify the CCD shift would fail in rare case when the oracle brings no improvement**. In this case, we cannot tell if CCD shift does not appear or our oracle is too weak to identify it. However, in our experiments this case does not appear, which means the oracle $P_{N_j,bal}(x|y_j)$ is usually much more reliable than CCD from long-tailed distribution, making our empirical study effective. We have added the discussion on **the failing cases** to the revised submission.
>
> >It is unclear if equation (4) is actually used to sample datasets in prior work.
>
> Our formulation of (4) may cause some misundrstanding. It is more accurate to describe (4) by "while instances in long-tailed datasets are sampled from".  (4) is indeed the empirical distribution of a long-tailed dataset as the reviewer points. The difference that (5) makes from (4) is the *balanced* empirical distribution $P_{N_j,bal}(x|y_j)$ as an oracle for real CCD, i.e. more samples of tail classes are drawn. (5) serves for the empirical study in Section 3.
>
> The most general sampling method from a long-tailed dataset is following (4) to sample each instance, named as "instance-balanced sampling" in [1].  To solve the long-tailed problem, some works [1,2,3] propose re-sampling methods to modify label distribution in which the samples can be repeated, e.g. DRS [2] assigns the same sampling probability $P(y)$ to each instance of class $y$ (the implementation is "ImbalancedDatasetSampler" in https://github.com/kaidic/LDAM-DRW/blob/master/utils.py). As we focus on CCD shift,  (5) does not modify the im-balanced label distribution.
>
> [1] Decoupling representation and classifier for long-tailed recognition. ICLR 2020.
> [2] Learning Imbalanced Datasets with Label-Distribution-Aware Margin Loss. NeurIPS 2019.
> [3] Class-Balanced Loss Based on Effective Number of Samples. CVPR 2019.

---

> > ### Comment · Reviewer_7ExN · 2022-11-19
> > **Reply to rebuttal**
> >
> > I thank the authors for their efforts with their rebuttal and detailed response. However, I am not sure if I understand your answer to my weaknesses 1-4. In particular, do the references [1,2,3] that authors pointed to use equation (5)? I am not satisfied with the experiments based on eq (5) because we may see a lot of repeated examples from long tail classes in the case of (4) than in the case of (5).
> >
> > >In our empirical study, with sample distribution (5), the results are improved significantly (Figure 1) and the counter-intuitive facts do not appear (Figure 2, Table 1), which suggests that CCD from long-tailed distribution is unreliable and CCD shift influences the performance a lot.
> >
> > This is also simply a finite-sample issue due to repeated examples in (4) but not in (5). Is it really long-tailed if one can sample novel examples but will the smaller frequency of sampling, i.e., if you actually count the number of unique examples throughout the training with (5), you may actually not see a tailed behavior?

---

> > > ### Author Response · Authors · 2022-11-22
> > > **More unique samples do not solve long-tailed label distribution**
> > >
> > > Thanks for your engagement. We really appreciate your response.
> > >
> > > We agree that sampling based on (5) makes the model see more novel (and unique) instances than based on (4). However, **it is the label distribution $p(y)$ (dependent on the frequency of sampling) instead of the number of unique samples that determines whether a distribution is long-tailed [2,4,5,6]. Even balanced number of unique samples does not mean the distribution is balanced.**
> > >
> > > In our empirical study, sampling based on (5) changes the class-conditional distribution $p(x|y)$ while keeping $p(y)$ unchanged. So we get a long-tailed distribution with removed (to some extent) CCD shift. Further, we can perform experiments on this distribution to identify the influence of CCD shift in long-tailed recognition.
> > >
> > > To put it more convincingly, following your constructive comments, we count the number of unique samples in our experiments. In the following table, the number of unique samples becomes more balanced as the sampling times ($\sharp$ epochs) increase. However, the label distribution is still imbalanced as indicated by $p(y)$.  Moreover, the performance of ERM with sampling from (5) still shows imbalance and the rebalancing method (e.g., DRW) can bring significant improvement(same trends on CIFAR100-LT, please refer to Figure 5 and Table 3).  The improvement is not from changing long-tailed $p(y)$ but changing $p(x|y)$, which is right the factor our study cares about. That is, more unique samples alleviate CCD shift but do not solve label shift problem i.e. long-tailed class distribution.
> > > |CIFAR10-LT| class 1 |   | | | | | | | | class L |
> > > |:------|:---------|:-----|:------|:------|:---------|:-----|:------|:------|:---------|:-----|
> > > |$N*P(y)$ (label distribution)|5000|2997|1796|1077|645|387|232|139|83|50|
> > > |$\sharp$ unique samples: 50 epochs|5000.0|5000.0|5000.0|5000.0|4982.3|4848.8|4411.7|3661.0|2654.2|1826.6|
> > > |$\sharp$ unique samples: 100 epochs|5000.0|5000.0|5000.0|5000.0|5000.0|4994.3|4938.5|4623.0|3873.7|2977.9|
> > > |$\sharp$ unique samples: 200 epochs|5000.0|5000.0|5000.0|5000.0|5000.0|5000.0|4998.9|4965.5|4746.7|4146.8|
> > > |Accuracy: sampling from (4)|0.97|0.984|0.846|0.784|0.788|0.622|0.694|0.629|0.467|0.534|
> > > |Accuracy: sampling from (5)|0.969|0.989|0.908|0.819|0.899|0.732|0.853|0.784|0.746|0.725|
> > > |Accuracy: sampling from (5) with DRW|0.912|0.948|0.861|0.768|0.901|0.814|0.923|0.904|0.928|0.922|
> > >
> > > (The count of unique samples is an average of 3 seeds.)
> > > [1,2,3] do not use (5) since (5) is proposed to remove CCD shift by us. Instead, they use resampling to change the label distribution $p(y)$ to be more balanced with the number of unique samples unchanged.
> > >
> > > [1] Decoupling representation and classifier for long-tailed recognition. ICLR 2020.
> > > [2] Learning Imbalanced Datasets with Label-Distribution-Aware Margin Loss. NeurIPS 2019.
> > > [3] Class-Balanced Loss Based on Effective Number of Samples. CVPR 2019.
> > > [4] Long-tail learning via logit adjustment. ICLR 2021.
> > > [5] Distribution Alignment: A Unified Framework for Long-tail Visual Recognition. CVPR 2021.
> > > [6] Optimal transport for long-tailed recognition with learnable cost matrix. ICLR 2022.
> > >
> > > Thanks again for your reply and detailed comments. If there are any remaining issues, please let us know, we would be happy to address them.

---

> > > > ### Comment · Reviewer_7ExN · 2022-11-28
> > > > **Thanks for your reply**
> > > >
> > > > > However, it is the label distribution  (dependent on the frequency of sampling) instead of the number of unique samples that determines whether a distribution is long-tailed [2,4,5,6].
> > > >
> > > > I agree with the author's comment that the label distribution determines whether a distribution is long-tailed. However, I am not sure that **"Even a balanced number of unique samples does not mean the distribution is balanced"** from the lens of overparameterized models when we have finite samples because essentially the model is seeing some repetitive examples. I imagine that the issues with sampling from (5) would be different from the issues with sampling from (4) for overparameterized models with finite samples.
> > > >
> > > > Moreover, I am not sure if eq (5) is also practical, in the sense that why would one use labeled data with less frequency if relatively more samples are available?

---

> > > > > ### Author Response · Authors · 2022-12-01
> > > > > **Thanks for your feedback**
> > > > >
> > > > > Thanks for your comments.
> > > > >
> > > > > **Resampling over balanced number of unique samples may lead to imbalanced label distribution $p(y)$**, so the number of unique samples alone cannot determine whether $p(y)$ is balance or not.
> > > > >
> > > > > We agree that
> > > > > > sampling from (5) would be different from the issues with sampling from (4) for overparameterized models with finite samples.
> > > > >
> > > > > The difference lies in where the CCD $p(x|y)$, i.e., long-tailed in (4) vs balanced in (5), whereas both (4) and (5) have the same long-tailed label distribution $p(y)$.  In other words, sampling with (5) can alleviate CCD shift (which (4) suffers from) and hence get less repeated samples, while keeping long-tailed label distribution $p(y)$ unchanged.
> > > > >
> > > > > As you concern, Eq. (5) is not practical at all. However, **we do not use (5) in our long-tailed learning process. Instead, we use (5) to identify the influence of CCD shift** （i.e., what if we alleviate the CCD shift by leveraging the balanced CCD via (5)).

---

> ### Author Response · Authors · 2022-11-15
> **Response to reviewer 7ExN(2/2)**
>
>
> >In Algorithm 1, it is unclear why the loop in Line 4 will be executed with $g_{i}^*$ initialized the way it is in Line 3.
>
> $||g_i^*||<\epsilon$  in Line 4 should be $||g_i^*||>\epsilon$.   ${g_i}^{\*}$ is the gradient in the inner-optimization, by which the original instance is transformed for data augmentation. The $||g_i^*||>\epsilon$ is the condition to continue gradient descent in the inner-optimization. That is, we search an $\epsilon$-stationary point, which serves for the convergence bound in Theorem 3 in the original paper (Eq. (10) or Theorem 5 in the revised paper). Thanks for pointing out the typo and we have revised the paper accordingly.
> >Figure 3 is also a bit unclear. It is hard to follow the description of the DRA method from this figure.
>
> The learning process of DRA involves inner-optimization and outer-optimization, which are indicated by orange and blue arrows respectively in Figure 3.
> - The inner-optimization generates augmentation examples, in which the left image is input to the network and output one or a sequence of examples as data augmentation, corresponding to "From the M-th loop" and "From the k-th,...,M-th loops" respectively.
> - The outer-optimization updates the networks using the augmented training instances (outputs from inner-optimization).
> - The curved orange and blue arrows denote gradient back-propagation in inner- and outer-optimization process respectively.
>
> >Missing related work:The assumption is well known as label shift.
>
> We appreciate the reviewer for the efforts to help us not miss potential related works. We will add some discussion on label shift problem with references the reviewer mentioned. The task of a typical label shift problem is detecting and estimating the label shift, and applying some strategy to address the label shift problem, e.g. reweighting and resampling as in the references recommended by the reviewer [4,5,6,7]. Long-tailed recognition considers a more specific setting where the label shift is known. In this paper, we focus our study on class-conditional distribution (CCD) shift besides label shift in long-tailed recognition context.
>
> Besides, the reviewer qfH1 also mentions a work utilizing DRO to solve label shift [8], which is orthogonal to our work. Please refer to our reply to reviewer qfH1 for more information.
>
> [4] Detecting and Correcting for Label Shift with Black Box Predictors. ICML 2018.
> [5] A unified view of label shift estimation. NeurIPS 2020.
> [6] When Training and Test Sets Are Different: Characterizing Learning Transfer. Dataset Shift in Machine Learning 2009.
> [7] Adjusting the Outputs of a Classifier to New Priori Probabilities: A Simple Procedure. Neural Computation 2002.
> [8] Coping with Label Shift via Distributionally Robust Optimisation. ICLR 2021.
> >The connection with the proposed motivation is a bit thin.
>
> **DRA validates our insight which is motivated by the identified CCD shift.** We restate that our motivation is the identified CCD shift. To address it, we propose insight that training models robust to CCD shift would benefit long-tailed recognition. The proposed DRA directly validates the insight. Our Theorem 7 says with some mild conditions for the real CCD P(x|y) and optimization (common in prior works), the risk on balanced label distribution with ideal CCD is paritially bounded by the objective of DRA, which means that  **DRA provides an effective solution to the CCD shift**. Performance improvements brought by DRA in the experiments agree with the theoretical results.
>
>
> >Writing suggestions
>
> We revise the paper according to the comments from reviewers including clarification improvement of Section 3,4, review of WRM and label shift works, and a clearer statement of our contributions and DRA algorithm. Please refer to the new version we submitted.

---

### Official Review · Reviewer_qfH1 · 2022-10-25

**Confidence:** 4
**Correctness:** 3
**Technical Novelty And Significance:** 3
**Empirical Novelty And Significance:** 3
**Recommendation:** 5

**Clarity, Quality, Novelty And Reproducibility:**

* Clarity: Poor
* Quality: Fair
* Novelty: Fair
* Reproducibility: Fair

**Strength And Weaknesses:**

Strength

* Good motivation and theoretically proved algorithm;

Weakness

* The theory about the algorithm is trivial and does not provide much insight into the progress of the field;
* The high-level framework of DRA is missing. The paper quickly dived into details, making it very hard to understand the big picture;
* The connection between each part is not clear. For example, Eq.5 seems to express the paper's central idea but is never referred to/used in the next parts. The connection between Eq.5 and the theorems is also not clearly stated.
* The performance improvement seems marginal.
* The authors only conducted experiments on CIFAR and Imagenet-LT, which are relatively small datasets. Would the authors like to run experiments on larger datasets, such as iNaturelist, to confirm the effectiveness of the authors' data augmentation scheme?
* The paper seems to use a sampling method to remove the CCD shift (Eq. 5), so the problem becomes a traditional label shift problem. Then it applies the DRO to solve the label shift problem. The second part is quite similar to the work "Coping with Label Shift via Distributionally Robust Optimisation, ICLR 2021". Could the author tell the difference between the second part of this paper and that paper?
* The paper proposed the sampling method (Eq.5) and then quickly moved to the optimization method. I'd like to see a more thorough analysis of the sampling method, including in what circumstances the sampling method is guaranteed to remove the CCD shift.


**Summary Of The Paper:**

The paper studied the phenomenon that inconsistent CCD causes bad performance in long-tailed recognition. It proposed a new approach DRA, for data augmentation. The theoretical properties of the proposed method were proven, and experimental results revealed the improvement of applying DRA in long-tailed problems.

**Summary Of The Review:**

The motivation of this paper is good, and the method seems reasonable. However, the theoretical analysis does not provide much insight into the field, and the performance improvement is marginal. Furthermore, the paper's organization is not good, making the paper very hard to read.

---

> ### Author Response · Authors · 2022-11-15
> **Response to reviewer qfH1(1/5)**
>
> We appreciate the careful and constructive comments from the reviewer and we hope our clarification will address your concerns.
>
> To begin with, we make a summary of our work for better clarification: we identify a rarely noticed but essential issue in long-tailed recognition, class-conditional distribution (CCD) shift. Motivated by our finding, we give a new insight of long-tailed recognition: training models robust to CCD shift would benefit long-tailed recognition. DRA is proposed to validate the insight by utilizing DRO technology with simple but effective modifications. The validation is supported by our analysis empirically and theoretically.
>
> >The theory about the algorithm is trivial and does not provide much insight into the progress of the field; (1/2)
>
> We argue that there are *misunderstandings of our contributions* on our theoretical analysis and DRA algorithm.
> 1. **Our theoretical results give useful insight to long-tailed learning, including training models robust to CCD shift is necessary for long-tailed recognition.**
>  Our Theorem 7 gives a novel and practical generalization bound as below for any choices of hyperparameters $\{\lambda_j\}_{j\in [L]}$:
>
>
> $$R_{bal,l}\leq \frac{1}{L} \sum_{j\in[L]} \overbrace{\lambda_{j}C(N_j)}^{generalization\ gap}+\overbrace{E_{P_{N_j,LT}}(x|y_{j})[\sup_{(x_{z},y_{z})}l(x_{z},y_{z})-\lambda_j c((x_{z},y_{z}),(x,y))]}^{training\ loss\ of\ DRA} $$
>
> where $R_{bal,l}$ is the risk on the balanced distribution with ideal CCD $P(x|y_j)$ and $C(N_j)$ is a constant for specific training sets and is decreasing function of $N_j$ i.e. the number of instances from class $j$.  The second term is actually the training objective of DRA in the (10) of Theorem 2. Therefore, the first term $\lambda_jC(N_j)$ serves as the generalization gap between training loss of DRA and the risk on the ideal balanced test distribution.
>
> The bound gives below insights to long-tailed recognition:
>   - **The robustness to CCD shift is necessary for the generalization of long-tailed learning**, as using empirical CCD leads to large generalization gap.  If we do not force robustness, i.e. the Lagrange multipiers $\lambda_j$ are extremely large, DRA actually equals to ERM . In this situation, large $\lambda_j$ leads to considerable generalization gap, especially for the tail classes with larger $C(N_j)$.
>   - **The expected robustness shall be assigned by classes to address CCD shift, i.e. each class needs different amount of robustness** corresponding to different reliability of the empirical distribution of each class. Therefore, to obtain a relatively small generalization gap, the multipiers should be smaller for the classes with fewer instances.
>   - **Greedily pursuing robustness would get sub-optimal performance**. The bound also reflects the trade-off between robustness and  performance. Greedily pursuing robustness, i.e. minimizing the generalization gap $\lambda_jC(N_j)$ by setting extremely small $\lambda_j$, will not guarantee a tighter bound since smaller $\lambda_j$ increases the objective of DRA. The most extreme situation is setting $\lambda$ to zero and it causes a trivial bound meaning the model refuses to make predictions, which is well-known as "over-pessimism" in DRO. Please refer to [1,3,4] for more clarification.
>
> The bound also shows a new viewpoint beyond existing theoretical results to the long-tailed learning community. Specifically, it
>   - is more tight compared to WRM. It considers all possible multipliers $\{\lambda_{j}\}_{j\in[L]}$ while WRM is a special case $\lambda_j=C, for\ all\ j \in [L]$.
>   - is closer to practical training compared to the Fisher-consistency result by logit adjustment. Fisher-consistency theory from Theorem 1 in [5] only states that a Bayes-optimal classifier would be obtained by the minimization of logit adjustment loss under a balanced label distribution with real CCDs.
>   - is no longer dependent on the capacity of the hypothesis class, which is extremely large for modern neural networks, while the bound from LDAM (Theorem 1 in [6]) is.
>
> [1] Certifying some distributional robustness with principled adversarial training. ICLR 2018.
> [2] Wasserstein distributioally robust optimization: Theory and applications in machine learning. INFORMS 2019.
> [3] Incorporating Unlabeled Data into Distributionally Robust Learning. JMLR 2021.
> [4] Does distributionally robust supervised learning give robust classifiers? ICLR 2018.
> [5] Long-tail learning via logit adjustment. ICLR 2021.
> [6] Learning imbalanced datasets with label-distribution-aware margin loss. NeurIPS 2019.

---

> ### Author Response · Authors · 2022-11-15
> **Response to reviewer qfH1(2/5)**
>
> >The theory about the algorithm is trivial and does not provide much insight into the progress of the field; (2/2)
>
> 2. **Our theoretical result inspires the modifications that DRA makes over existing method, which are completely non-trivial** and verified by the empirical results.
> The proposed DRA is based on prior DRO method WRM [1] with two main modifications: assigning class-wise radius to the uncertainty set and using a sequence of examples as data augmentation instead of the last one.
> Firstly, we propose class-aware uncertainty set which is a novel and reasonable solution to address CCD shift. Directly adapting WRM to class-aware uncertainty set cannot arrive a tractable min-max problem, while **our Theorem 2 guarantees that DRA with class-aware uncertainty set can be converted to a min-max problem**. (If a theorem certificates the feasibility of a new strategy to a newly identified issue, it cannot be trivial.) Secondly, to solve the convergence and instable issue of WRM in computing the objective (10), we propose using a sequence of examples as data augmentation. **Theorem 3 gives an evidence that this new strategy contributes to more stable optimization.** The significant empirical improvement brought by DRA over original WRM (in Figure 4, Table 8) verifies that our theoretical results and DRA algorithm are valuable and effective.
>
> [1] Certifying some distributional robustness with principled adversarial training. ICLR 2018.
> [2] Wasserstein distributioally robust optimization: Theory and applications in machine learning. INFORMS 2019.
> [3] Incorporating Unlabeled Data into Distributionally Robust Learning. JMLR 2021.
> [4] Does distributionally robust supervised learning give robust classifiers? ICLR 2018.
> [5] Long-tail learning via logit adjustment. ICLR 2021.
> [6] Learning imbalanced datasets with label-distribution-aware margin loss. NeurIPS 2019.

---

> ### Author Response · Authors · 2022-11-15
> **Response to reviewer qfH1(3/5)**
>
> >Concerns about (5): the 3th, 6th and 7th weaknesses
>
> There exist some misunderstandings about (5) and our proposed method. We make clarifications as follows:
> 1. **The (5) is not a part of our proposed method, but serves as an oracle to the ideal CCD $P(x|y)$ in the empirical study to explain the influence of CCD shift.**
> To identify the CCD shift as a key factor affecting long-tailed recognition, (5) is used as an oracle to estimate the real or "ideal" CCD to make comparison with the original empirical CCD in long-tailed distribution. Specifically, assuming the CCD from long-tailed distribution $P_{N_j,LT}$ has been relatively reliable i.e. the left term of Proposition 1, $E_{P(x|y_j)}\{l(x,y)\}-E_{P_{N_j,LT}}\{l(x,y)\}$ is small, then using (5) to get more reliable CCD $P_{N_j,bal}(x|y_j)$ would not give significant improvement. As in Proposition 1, for a already small left term, a tighter bound with larger $N_j$ will not help. But the experiments (Figure 1,5, Table 3) show sampling from (5) improves a lot for long-tailed recognition, which proves from the reverse that CCD from long-tailed distribution is unreliable and significantly affects the model performance.
> On the other hand, we agree that there hardly exists a guarantee that the (5) can remove CCD shift, since how much $P_{N_j,bal}$ in (5) approximates the ideal CCD $P(x|y)$ cannot be computed. However, using (5) is valid for our study to identify CCD shift as **we only need to pursue how unreliable the CCD from long-tailed recognition is, instead of how good the CCD from balanced training set is**. For more clarification please refer to the current submission section 3.2 or our reply to reviewer 7ExN .
> 1. **We do not think that long-tailed recognition is a traditional label shift problem.** The task of a traditional label shift problem is detecting and estimating the label shift, and applying some strategy to address the label shift problem, e.g. reweighting and resampling. Long-tailed recognition considers a more specific setting where the label shift is known.
> However, only solving label shift is not enough for long-tailed recognition. **Our work target at solving CCD shift besides label shift**, i.e. assuming $P_{train}(y)\neq P_{test}(y)$ and $P_{train}(x|y)\neq P_{test}(x|y)$. After identifying the CCD shift in Section 3, our work gives a novel insight and then proposes a feasible algorithm for long-tailed recognition.
> 1. **Sampling according to (5) cannot be applied to remove the CCD shift in training** but to facilitate the discovering of the influence from CCD shift.  In fact, (5) is not avaiable for long-tailed training data and removing CCD shift from scarce data from the tail classes is intractable.  Instead, we propose a new and more feasible idea for long-tailed recognition: training models robust to CCD shift, whose effectiveness is validated by the proposed DRA theoretically and empirically.

---

> ### Author Response · Authors · 2022-11-15
> **Response to reviewer qfH1(4/5)**
>
> >Discussion with "Coping with Label Shift via Distributionally Robust Optimisation[7]"
>
> We thank the reviewer again for reminding us of this related work. We think the discussion on [7] could better demonstrate the superiority of our DRA and provides useful insights. We have revised the paper with the discussion about this work accordingly.
>
> The proposed method AdvShift by [7] uses DRO to solve label shift i.e. training a model robust to potential label distribution besides the training label distribution, whose objective is:
> $$\min_{\theta}\sup_{\pi\in\mathcal{P}}E_{\pi}[l_{\theta}(x,y)],\ \mathcal{P}=\{\pi\in \Delta^{[L]}|d(\pi,p_{emp})<r\}$$
> where $\pi$ is the label distribution and $\mathcal{P}$ is the uncertainty set of potential label distributions that the model need to be robust to.
>
> Comparing with our objective in (7), we claim that
> 1. On the long-tailed recognition task, AdvShift serves to solve the shift of labels $P(y)$, while DRA is proposed to train a model robust to shift of CCD $P(x|y)$. Therefore, **[7] is parallel to many previous works which solve the same label shift problem, e.g. re-weighting, re-sampling, logit adjustment [5], etc. and orthogonal to our work**. The motivation and method of the two works are different and hardly overlapped.
>
> 2. Although long-tailed recognition could be seen as a special label shift problem with known test label distribution, AdvShift may not be an appropriate solution for long-tailed recognition.  On the one hand, **AdvShift demands unnecessary robustness for long-tailed recognition i.e. robustness to any label distribution near the training label distribution** However, the test label distribution is known under general setting and usually balanced in long-tailed recognition. This is to say, AdvShift has to set the training label distribution as the center of uncertainty set. Consequently, when the training label distribution is extremely im-balanced (i.e. the KL-distance between the training label distribution and balanced test label distribution is extremely large),  AdvShift demands an extremely large radius of the uncertainty set, which may bring a well-known issue "over-pessimism"[3,4] in DRO. On the other hand, AdvShift does care the CCD shift. It generates adversarial label distribution and estimates the risk of each class by empirical CCD, so **the CCD shift issue still exists in AdvShift**, which inevitably limits its performance.
>
> [7] Coping with Label Shift via Distributionally Robust Optimisation. ICLR 2021.

---

> ### Author Response · Authors · 2022-11-15
> **Response to reviewer qfH1(5/5)**
>
> >Concerns of empirical performance and experiments: the 4th and 5th weaknesses
> 1. DRA can **consistently improve various strong baselines** on all datasets in our experiments. Our experiments prove that DRA is suitable to main kinds of methods in long-tailed recognition, including decoupling methods (e.g. CRT[8]), resampling methods (e.g. DRS[9]),  post-hoc adjustment (e.g. PC softmax[10], logit adjustment[5]), which are relatively strong baselines (previous state-of-the-arts). Suitability to various kinds of such strong baselines is a big superiority as previous works usually report improvement on only one kind of baselines [10,11,12].
> 2. **The improvement DRA brings is more significant than previous works**. As baselines we choose are relatively strong, obtaining improvement is not that easy. In the following table, we compare the improvement of DRA with recent works LADE and GIT, which are based on different baselines including logit adjustment, DRS. We can see DRA can improve the performance more significantly.
>
> | dataset |  baseline | result of baseline| previous work | result of previous work | DRA |
> |:------|:---------|:-----|:------|:------|:------|
> | CIFAR10-LT|logit adjustment| 79.65| LADE[9]| +0.12   | +0.68 |
> | CIFAR100-LT|logit adjustment| 45.17| LADE| +0.22   | +0.54 |
> | ImageNet-LT(90 epochs)      |logit adjustment|49.25 |LADE|-0.05|+0.24|
> | ImageNet-LT(180 epochs)      |logit adjustment|51.90 |LADE|+0.24|+0.40|
> | CIFAR10-LT*|CE-DRS| 75.87| GIT[11]| +1.19   | +1.51 |
> | CIFAR100-LT*|CE-DRS| 41.21| GIT| +0.65   | +1.40 |
> |CIFAR10-LT*|LDAM-DRS| 77.47| GIT| +1.02   | +1.29 |
> | CIFAR100-LT*|LDAM-DRS| 42.78| GIT| +0.71   | +0.75 |
>
>  \* means using batchsize 128 because we find the official code of GIT cannot work well under batchsize of 256. Therefore, despite DRA cannot beat all SOTA methods, it brings considerable improvements and surpasses the improvements of previous works on the same baselines. In additional, DRA does not have extra requirements on training and model structure (while some previous works do, e.g. GIT[12], M2m[13]).
> In summary, **the empirical improvement of DRA is non-trivial**.
>
> 3. We are also happy to add experiments on more datasets, e.g. iNaturalist, to make more convincing results. However, there is not enough time for finishing our experiments within the discussion stage. We will add these new experiments to the paper as soon as they have been done. As a remedy, we produce more results on ImageNet-LT which is a relatively large-scale dataset as well, e.g, logit adjustment-DRA with various epochs, to make the results of DRA more convincing. We wish our clarification and empirical results could address your concerns.
>
> [8] Decoupling representation and classifier for long-tailed recognition. ICLR 2020.
> [9] Learning imbalanced datasets with label-distribution-aware margin loss. NeurIPS 2019.
> [10] Disentangling label distribution for long-tailed visual recognition. CVPR 2021.
> [11] Improving Calibration for Long-Tailed Recognition. CVPR 2021.
> [12] Do deep networks transfer invariances across classes? ICLR 2022.
> [13] M2m: Imbalanced classification via major-to-minor translation. CVPR 2020.
>
> >The high-level framework of DRA is missing. The paper quickly dived into details, making it very hard to understand the big picture;
>
> Thanks for pointing out these presentation issues. In the revised paper, we clear up the sampling method in Section 3, add more description on the high-level framework of DRA in Section 4 and provide clarifications for all above misunderstandings. Please refer to the new version we submitted.

---

### Official Review · Reviewer_Lnfc · 2022-10-30

**Confidence:** 5
**Correctness:** 3
**Technical Novelty And Significance:** 2
**Empirical Novelty And Significance:** 2
**Recommendation:** 3

**Clarity, Quality, Novelty And Reproducibility:**

For clarity, quality, and novelty, see Strength And Weaknesses;
Reproducibility should be okay.

**Strength And Weaknesses:**

# Strength
1. The studied problem is interesting and not well-addressed in existing literature;
2. The proposed method is well-motivated and equipped with theoretical analyses;
3. Extensive experiments show the advantages of the proposed method;
4. This writing is clear and easy to follow.

# Weakness
1. Novelty: the proposed Distributionally Robust Augmentation (DRA) approach is borrowed from previous work DRO and no significant improvement has been made. The used min-max optimization is very commonly used in related works;
2. Empirical improvement is weak. As reported in Table 2 and 7, the proposed DRA does not consistently outperforms existing ones, such as Logit Adjustment, PC softmax, LADE;
3. Rationale behind the approach: There is a gap between the DRA and CCD shift. Why DRA can remove CCD shift? How accurate can DRA approximate real CCD? Those questions seem not answered in the theorems.
4. Hyperparameters: the method involves several hyperparameters, e.g., $C, S, \beta, M, \alpha_{inner}$ and some of them require a validation set for tuning which is not applicable in some cases.

**Summary Of The Paper:**

This paper investigates the Class-Conditional Distribution (CCD) shift issue in long-tailed recognition due to scarce instances, which exhibits a significant discrepancy between the empirical CCDs for training and test data, especially for tail classes. To alleviate the issue, this paper presents a data augmentation approach to generate more virtual samples.

**Summary Of The Review:**

This paper studies a relatively new problem in long-tailed learning and a simple strategy is proposed. However, the proposed method seems not novel enough and has several hyperparameters. In addition, experimental show that the advantages of the method is not significant.

---

> ### Author Response · Authors · 2022-11-15
> **Response to reviewer Lnfc(1/2)**
>
> We thank the reviewer for the comments and suggestions.
> > Novelty: DRA is borrowed from previous work DRO and no significant improvement has been made. The used min-max optimization is very commonly used in related works.
>
> We cannot agree that the proposed DRA is simply borrowed from previous method without significant improvement. In fact, previous DRO method WRM [1] makes very little empirical improvement on long-tailed recognition (please refer to Table 8, Figure 4 and the clarification below). Besides, we do not claim the min-max optimization is the main contribution of this work.
>
> The novelty and effectiveness of DRA can be explained from theoretical and empirical aspects respectively.
>
> - Theoretically, the novel points of DRA compared to related DRO method (WRM) include the class-aware uncertainty set radius and a sequence of examples in data augmentation.
>   - Class-aware uncertainty set is a novel and reasonable solution to address CCD shift, which assigns class-wise robustness corresponding to the varying reliability across different empirical distributions. The result of WRM cannot arrive at a tractable min-max problem under our uncertainty set, while our Theorem 2(in the original version, Theorem 4 in the current version) guarantees that DRA with class-aware uncertainty set can be converted to a min-max problem.  Moreover, the class-aware uncertainty set admits the generalization bound in the Theorem 7, which gives useful insight to long-tailed learning, including training models robust to CCD shift is necessary for long-tailed recognition. Please refer to our reply to reviewer qfH1 for more clarification.
>   - We propose to use a sequence of examples as data augmentation to solve the convergence and instability issue that WRM has in computing the objective similar to (10) with large multipliers. Our Theorem 3(in the original version, Theorem 5 in the current version) indicates that this new strategy could bring more stable optimization result under mild conditions similar to WRM.
> - Empirical study of DRA verifies its effectiveness. As in Figure 4, class-aware radius exhibits significant superiority over WRM, and using a sequence of augmentation examples obtains even better and more stable results than using a single example.
>
> In summary, we claim that **DRA is not simply applying existing DRO technology to long-tailed recognition, in contrast, it is novel and insightful**.
>
> Please also be noted that DRA is only one of our main contributions in this work, which are restated as follows:
> 1. We identify a rarely noticed but essential issue in long-tailed recognition, class-conditional distribution (CCD) shift.
> 2. We make and validate new insights to counter-intuitive facts of previous methods from CCD shift view, e.g. why re-balancing strategies generate sub-optimal features.
> 3. We theoretically discover that training models robust to CCD shift would benefit long-tailed recognition.
> 4. We propose DRA inspired by the theoretical results, and prove its effectiveness theoretically and empirically .
>
> References:
> [1] Certifying some distributional robustness with principled adversarial training. ICLR 2018.
>
> > Empirical improvement is weak. DRA does not consistently outperform existing ones, such as Logit Adjustment, PC softmax, LADE;
>
>  Although the improvement DRA brings does not beat all state-of-the-art methods on the four datasets, the statement that “*empirical improvement is weak*” is NOT the truth. In fact, DRA exhibits consistent and significant improvement:
> 1. DRA brings improvement to all baselines considered in our experiments, including deferred re-balancing (e.g. CE-DRS, LDAM-DRS [2]), decoupling methods (e.g. CRT [3]), Logit Adjustment [4],  and PC softmax [5]. Results in Table 2 and 7 verify this.
> 2. Based on new results we produced, DRA improves strong baseline PC softmax [5] and Logit Adjustment [4] and beats LADE [5] on both CIFAR-LT and ImageNet-LT datasets, as shown in Table 2 and 7 in the new submission.
>
> References:
> [2] Learning imbalanced datasets with label-distribution-aware margin loss. NeurIPS 2019.
> [3] Decoupling representation and classifier for long-tailed recognition. ICLR 2020.
> [4] Long-tail learning via logit adjustment. ICLR 2021.
> [5] Disentangling label distribution for long-tailed visual recognition. CVPR 2021.

---

> ### Author Response · Authors · 2022-11-15
> **Response to reviewer Lnfc(2/2)**
>
> > There is a gap between the DRA and CCD shift. Why DRA can remove CCD shift? How accurately can DRA approximate real CCD? Those questions seem not answered in the theorems.
>
> There should be some misunderstanding. We do not aim at removing CCD shift, i.e. restoring real CCD, which is intractable with very scarce instances. Instead, we propose the insight that training models robust to CCD shift would benefit long-tailed recognition and validate it by DRA.  Theorem 7 gives a generalization guarantee under mild conditions, showing DRA is a principled solution to the long-tailed recognition problem.
>
> > Hyperparameters: the method involves several hyperparameters, e.g.$C,S,\beta,M,\alpha_{inner}$,  and some of them require a validation set for tuning which is not applicable in some cases.
>
> First we should emphasize that according to the "no free-lunch theorem"[6], **it is impossible to completely avoid hyperparameters**, since a model needs different hyperparameters to adapt to different working scenarios.
>
> The general setting of long-tailed recognition allows a validation set separated from training dataset. Many previous related works also need adjusting hyperparameters on the validation set such as M2m[7], LADE[5], GIT[8].  That is, **existence of hyperparameters is not a big issue as long as they are not very sensitive and easy to set.**
>
> In our work, the hyperparameters possess such merits. In our experiments, C, $a_{Inner}$, M works well under the default values provided by us. We list them as hyperparameters just to remind users that these parameters can be adjusted for better performance for specific tasks and datasets. DRA only needs to adjust three hyperparameters: k decides the number of augmentation samples,  $beta$ and S adjust the multiplier. There are only two options for k: M-1 or M/2. **Only $\beta$ and S need tunning on the validation set**. Moreover, as they have statistical meanings (which determine the Lagrangian multipliers corresponding to the uncertainty set constraints, representing the potential CCD shift of each class), it is easy to save computation from grid search.
>
> References:
> [5] Disentangling label distribution for long-tailed visual recognition. CVPR 2021.
> [6] No free lunch theorems for optimization. IEEE Transactions on Evolutionary Computation 1997.
> [7] M2m: Imbalanced classification via major-to-minor translation. CVPR 2020.
> [8] Do deep networks transfer invariances across classes? ICLR 2022.

---

### Author Response · Authors · 2022-11-15
**Response to all reviewers**

We would like to thank all of the reviewers for their careful reviews and feedbacks, which help us to improve the quality of this paper.

According to these constructive comments, we have revised our manuscript to clarify some misunderstandings and added more experiments to make our results more convincing. Specifically, we
- improve the presentation and clarification of DRA in Section 4, including the objective reformulation, the high-level framework aligned with our illustration, and the learning process of DRA. (**qfH1, 7ExN**)
- clarify Proposition 1 and the sampling method (5) in empirical study based on which we  identify CCD shift. (**qfH1, 7ExN**)
- supply more results of DRA on large-scale dataset ImageNet-LT in Table 7, which verify that our DRA consistently beats prior SOTA LADE. (**Lnfc, qfH1, 9ohR**)
- give remarks for our generalization bound in Theorem 7, including the superiority of ours over previous theoretical results and useful insights to long-tailed recognition. (**Lnfc, qfH1**)
- state our contributions more specifically in the abstract and introduction.(**7ExN**)
- add related works of label shift problem. (**qfH1, 7ExN**)
- add more details of WRM including its objective and algorithm, on which our DRA is based. (**qfH1**)

Besides these modifications to the manuscript, we also try our best to address the concerns of reviewers and clarify all potential misunderstandings. We hope all our efforts could address these concerns and welcome additional comments from reviewers to further improve this paper.

---

> ### Author Response · Authors · 2022-11-18
> **Thank you for reviewing**
>
> Dear reviewers and AC,
>
> we have put a lot of effort into responding to the concerns and comments from reviewers and improving our paper. We hope that our responses satisfactorily answer all the points raised and lead to an update of the scores. We also wonder if there are any other comments to improve our paper further. Unfortunately, we have not yet received any response from the reviewers. As the discussion period is drawing to close, we would appreciate the reviewers' feedback on our rebuttal.
>
> Best the authors

---

### Decision · Program_Chairs · 2023-01-20

**Decision:**

Reject

**Justification For Why Not Higher Score:**

All four reviewers recommend reject. Primarily, the paper fails to clarify the contribution compared to previous work on DRO, and uses a  confusing workflow using repeated unbalanced sampling from balanced data.

**Justification For Why Not Lower Score:**

N/A

**Metareview: Summary, Strengths And Weaknesses:**

This paper studies distribution shift of tail-classes in the learning setup of long-tailed recognition. Tail classes are more prone to such shifts. The paper proposes a data augmentation approach based on distributional-robust-optimization (DRO). The methods is compared with strong baselines, and yields small and consistent improvement on CIFAR-LT and Imagenet-LT.

Reviewers were concerned that the contribution should be more clear compared to previous work using DRO for LT and label shift. For instance, class-aware uncertainty and radius was studied in DRO-LT (Samuel et al), which also reaches a tractable min max problem. Also, there is a large number of papers about augmenting tail classes for improving long-tail. More stress should be given to the  methodological contribution, even with good empirical improvement.

There was some confusion regarding the case of resampling unbalanced data from a balanced distribution, and the difference between the sampling procedures of Eq 4. and Eq 5. The authors clarified in their rebuttal that Eq 5 is not used during LT learning.  At the end of the discussion, it should be clarified better how Eq 5 and E4 are used, for analyzing the CCD shift and for learning.  Some of the issues may be clarified if experiments can be added for data that is naturally long-tail (like Inaturalist), rather than artificially unbalancing data from a balanced dataset (like CIFAR-LT and Imagenet-LT). After all, that is the real scenario that the method aims to solve.

All four reviewers felt that the submitted paper was not ready for publication.  The authors submitted a rebuttal that provided answers to some reviewer concerns. But with the large number of issues that all reviewers raised, a revised version of the paper should go through a proper review process. I recommend that a revised version is submitted to another venue.


**Summary Of Ac-Reviewer Meeting:**

N/A